# LayerT2V: A Unified Multi-Layer Video Generation Framework

Guangzhao Li [1 2 *]  Kangrui Cen [3 4 *]  Baixuan Zhao [1]  Yi Xin [5 2]  Siqi Luo [1]  Guangtao Zhai [1]  Lei Zhang [3 4]
Xiaohong Liu [✉ 1 2]

## Abstract

Text-to-video generation has advanced rapidly, but existing methods typically output only the final composited video and lack editable layered representations, limiting their use in professional workflows. We propose **LayerT2V**, a unified multi-layer video generation framework that produces multiple semantically consistent outputs in a single inference pass: the full video, an independent background layer, and multiple foreground RGB layers with corresponding alpha mattes. Our key insight is that recent video generation backbones use high compression in both time and space, enabling us to serialize multiple layer representations along the temporal dimension and jointly model them on a shared generation trajectory. This turns cross-layer consistency into an intrinsic objective, improving semantic alignment and temporal coherence. To mitigate layer ambiguity and conditional leakage, we augment a shared DiT backbone with LayerAdaLN and layer-aware cross-attention modulation. LayerT2V is trained in three stages: alpha mask VAE adaptation, joint multi-layer learning, and multi-foreground extension. We also introduce **VidLayer**, the first large-scale dataset for multi-layer video generation. Extensive experiments demonstrate that LayerT2V substantially outperforms prior methods in visual fidelity, temporal consistency, and cross-layer coherence. To facilitate future research, we will release the code and dataset upon publication.

## 1. Introduction

In recent years, diffusion-based text-to-video (T2V) generation has made remarkable progress. Models such as Sora (OpenAI, 2024), Wan (Wan et al., 2025), and HunyuanVideo (Kong et al., 2024) can synthesize high-quality videos with complex motion and rich visual details from text prompts. However, the prevailing paradigm treats a video as a single, complete result rather than a collection of semantically separable layers, limiting its applicability in professional production scenarios that require precise control and local edits.

In real-world production pipelines, videos are rarely edited as a single block. Instead, creators rely on layered representations where foreground, background, and alpha matte are handled separately and then composed together, enabling flexible edits such as replacing backgrounds, refining subjects, or applying localized effects. Existing T2V models (Yang et al., 2024; Wan et al., 2025; Kong et al., 2024; OpenAI, 2024) output only the final composited video without explicit layer decomposition (*e.g.*, foreground, background, and alpha), leaving subsequent compositing and localized editing without a direct manipulation space.

Prior studies have explored layered generation under restricted settings, mostly focusing on producing a single RGBA foreground for images (Zhang & Agrawala, 2024; Quattrini et al., 2024) or videos (Dong et al., 2025), while lacking explicit background modeling and cross-layer consistency constraints. LayerFlow (Ji et al., 2025) investigates multi-layer joint generation, but due to limited high-quality data and the absence of explicit hierarchical interaction modeling, its results still suffer from limited stability and cross-layer coherence. Developing a unified architecture that produces high-quality, cross-layer consistent, and editable multi-layer videos remains a central challenge.

To address this, we propose **LayerT2V**, a unified framework that generates multi-layer videos in a single denoising process. Given text prompts, LayerT2V simultaneously produces multiple semantically consistent outputs: the composited video, an independent background layer, and one or more foreground RGB layers with their corresponding alpha mattes. Our key insight is that recent video diffusion models use very high compression in both time and

---
*Equal contribution [1]Shanghai Jiao Tong University [2]Shanghai Innovation Institute [3]Hong Kong Polytechnic University [4]OPPO Research Institute [5]Nanjing University. Correspondence to: Xiaohong Liu <xiaohongliu@sjtu.edu.cn>.

*Proceedings of the $43^{rd}$ International Conference on Machine Learning*, Seoul, South Korea. PMLR 306, 2026. Copyright 2026 by the author(s).

space (Blattmann et al., 2023; Wan et al., 2025), making it feasible to jointly model multiple layers along a shared denoising trajectory. This turns cross-layer consistency from an external post-processing constraint into an intrinsic generation objective.

However, directly extending existing architectures to the multi-layer setting introduces new challenges: unified generation can cause discontinuities near inter-layer boundaries, and different layers exhibit substantially different statistics, such as alpha mattes that are sparse and near-binary, fundamentally different from content-rich RGB layers. To address these issues, we introduce **LayerAdaLN** for layer-specific feature modulation, and **LayeredCrossAttention** to control layer-wise interactions with text conditions. Additionally, we finetune the Wan VAE (Wan et al., 2025) for alpha mask processing to improve matte reconstruction and stabilize multi-layer generation.

High-quality training data is a major bottleneck for multi-layer video generation. While recent large-scale video-text datasets have fueled T2V training (Bain et al., 2021; Chen et al., 2024; Wang et al., 2023; Nan et al., 2025), they do not provide layer-aligned supervision. We develop an automated pipeline for multi-layer data construction and cleaning, and build **VidLayer**, the first large-scale multi-layer video dataset containing approximately 4M frames. Each sample includes the full video, background layer, foreground layer, alpha matte, and fine-grained layer-level text descriptions. VidLayer fills the gap of high-quality multi-layer video data and provides a scalable, controllable, and evaluable foundation for multi-layer video generation.

In summary, our main contributions are:

- We propose **LayerT2V**, a unified framework that simultaneously produces multiple semantically consistent layer representations in a single inference pass.

- We construct **VidLayer**, the first large-scale multi-layer video dataset, providing a scalable and evaluable data foundation for multi-layer video generation.

- We introduce **LayerAdaLN** and **LayeredCrossAttention** to enable explicit layer modeling within a shared video diffusion backbone.

- Extensive experiments demonstrate that LayerT2V significantly outperforms prior methods in visual fidelity, temporal coherence, and cross-layer consistency.

## 2. Related Work

### 2.1. Controllable Video Generation

Video generation has undergone a paradigm shift from adapting 2D architectures to developing native 3D generative frameworks. Early approaches such as Make-A-Video (Singer et al., 2023), Tune-A-Video (Wu et al., 2022), as well as AnimateDiff (Guo et al., 2023) and VideoCrafter (Chen et al., 2023), inflated pretrained T2I backbones (U-Nets/LDMs) with temporal modules (*e.g.*, temporal / cross-frame attention), leveraging static visual priors but often facing temporal artifacts and limited long-horizon coherence. More recently, Diffusion Transformer (Peebles & Xie, 2023) based video generators have emerged, including VDT (Lu et al., 2024), Latte (Ma et al., 2024), and trajectory-aware variants such as Tora (Zhang et al., 2024). Modern DiT-style systems like CogVideoX (Yang et al., 2024), Wan (Wan et al., 2025), and Hunyuan (Kong et al., 2024) treat video latents as spatiotemporal patches with 3D VAEs to model complex motion and details, while Lumiere (Bar-Tal et al., 2024) introduced a Space-Time U-Net for full-frame-rate generation. Despite impressive visual quality, these models render foregrounds, backgrounds, and effects into a flattened RGB stream, lacking structural disentanglement and forcing users to regenerate entire scenes for minor edits—unsuitable for professional compositing workflows requiring layer-wise manipulation.

### 2.2. Layered Content Generation

Early studies on layered content generation focused on images. Text2Layer (Zhang et al., 2023) and LayerDiffuse (Zhang & Agrawala, 2024) pioneered text-driven RGBA synthesis by modeling transparency, while Alfie (Quattrini et al., 2024) reduced training costs via inference-time attention optimization. Extending to videos introduces temporal coherence challenges. TransPixeler (Wang et al., 2025), TransAnimate (Chen et al., 2025), and Wan-Alpha (Dong et al., 2025) address this through dedicated alpha tokens, motion modules, and distribution-centric approaches, respectively. However, these methods generate isolated transparent elements rather than coherent multi-layer videos. Recent efforts such as StM (Kara et al., 2025) and Over++ (Qi et al., 2025) study layered video composition by explicitly composing foreground and background layers. LayerFlow (Ji et al., 2025) is most related, serializing multiple layers along the temporal dimension, but often produces visual disconnection due to data scarcity and lack of explicit inter-layer consistency modeling.

## 3. Method

This section presents the overall design and training strategy of LayerT2V. We first review the background of Flow Matching in Section 3.1. Then, Section 3.2 introduces our unified multi-layer generation pipeline. Finally, Section 3.3 details our three-stage training strategy.

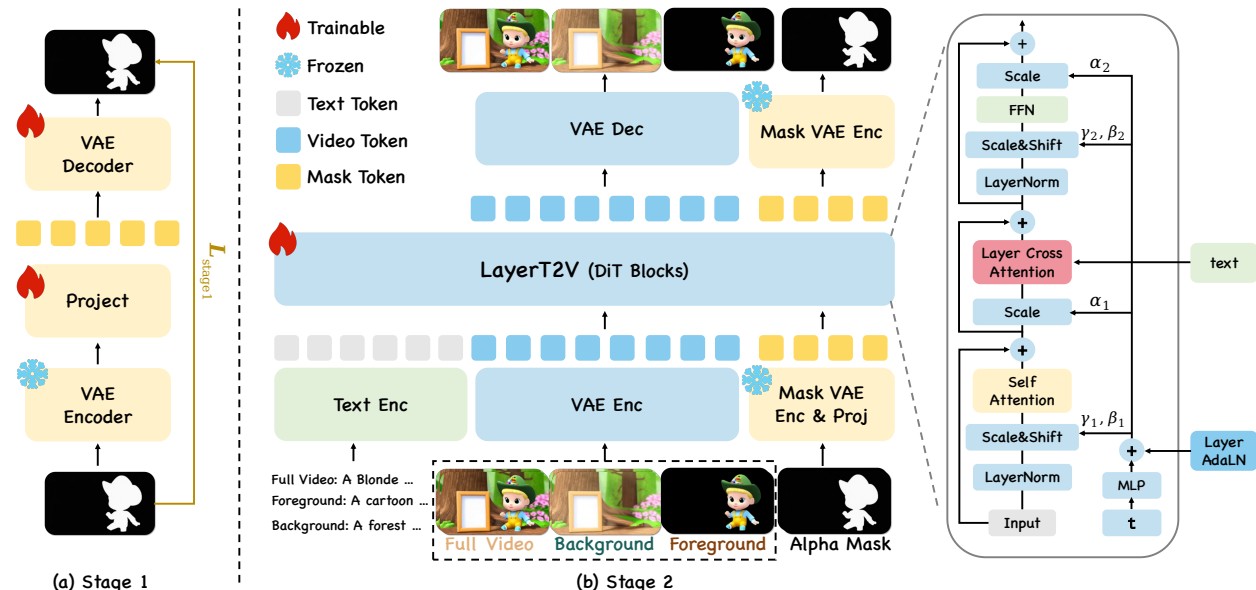

Figure 1. **Training pipeline and architecture of LayerT2V.** (a) Stage 1: Mask VAE adaptation, where the pretrained VAE encoder is frozen and a lightweight projection plus VAE decoder is trained to reconstruct high-quality alpha mattes. (b) Stage 2: Multi-layer generation with a DiT backbone that jointly models text tokens, video tokens, and mask tokens to generate Full Video, Background, Foreground, and Alpha Mask. LayerAdaLN injects layer identity into the timestep modulation, and Layered Cross-Attention conditions each layer on its corresponding text prompt to improve layer separation and cross-layer coherence.

### 3.1. Preliminary

**Flow Matching.** Flow Matching (Lipman et al., 2022) learns a time-dependent vector field that transports a prior $p_1$ (*e.g.*, $\mathcal{N}(0, I)$) to the data distribution $p_0$ by minimizing:

$$\mathcal{L}_{\text{FM}} = \mathbb{E}\big[ \|v_\theta(x, t) - v_t(x \,|\, x_0, x_1)\|^2 \big], \qquad (1)$$

where $t \sim \mathcal{U}(0, 1)$, $x_0 \sim p_0$, $x_1 \sim p_1$, $x \sim p_t(\cdot \,|\, x_0, x_1)$, and $v_t(x \,|\, x_0, x_1)$ is the oracle velocity along a prescribed path.

**Rectified Flow.** Rectified Flow (Liu et al., 2023) uses linear paths $x_t = (1 - t)x_0 + tx_1$ and learns:

$$\mathcal{L}_{\text{RF}} = \mathbb{E}\big[ \|v_\theta(x_t, t) - (x_1 - x_0)\|^2 \big], \qquad (2)$$

which simplifies the oracle velocity and enables efficient ODE integration.

### 3.2. LayerT2V Pipeline

As shown in Figure 1, LayerT2V jointly models multiple layers within a single generation process. While our framework naturally supports multiple foreground subjects (detailed in Stage 3 of Section 3.3), we present the single-foreground case here for clarity. We first encode each layer video (resolution $H \times W$, $T$ frames) into latent space via a VAE, obtaining latent representations $z_{\text{full}}$, $z_{\text{bg}}$, $z_{\text{fg}}$, and $z_{\text{mask}} \in \mathbb{R}^{C' \times T' \times H' \times W'}$, where $C'$, $T'$, $H'$, and $W'$ denote the channel, temporal, and spatial dimensions in latent space, respectively. We represent the foreground layer as

premultiplied content by multiplying the RGB video with the alpha mask before VAE encoding:

$$V_{\text{fg}} = V_{\text{full}} \odot A, \quad z_{\text{fg}} = E(V_{\text{fg}}), \qquad (3)$$

where $V_{\text{full}}$ is the full video, $A$ is the alpha mask, and $\odot$ denotes element-wise multiplication with channel broadcasting. We adopt the Wan VAE for RGB layers. However, alpha masks are single-channel, sparse, and near-binary, with statistics that differ substantially from natural RGB videos. Directly applying an RGB-pretrained VAE leads to degraded mask representations and may interfere with joint learning. We therefore fine-tune the Wan VAE to support masks (see Section 3.3). To construct a unified input for joint generation, we concatenate the latent codes along the temporal dimension:

$$z_0 := \text{Concat}\big([z_{\text{full}}, z_{\text{bg}}, z_{\text{fg}}, z_{\text{mask}}]\big), \qquad (4)$$

where concatenation is performed along the temporal dimension, $z_0 \in \mathbb{R}^{C' \times 4T' \times H' \times W'}$. Temporal concatenation preserves the input structure expected by pretrained video generators, enabling reuse of their temporal modeling capacity without modifying the architecture. Sampling a shared noise $z_1 \sim \mathcal{N}(0, I)$ encourages all layers to evolve consistently, improving cross-layer coherence.

Despite these advantages, temporal serialization also introduces a new ambiguity: the backbone must disentangle intrinsic temporal dynamics from the artificial layer ordering. Without explicit layer cues, tokens from different layers

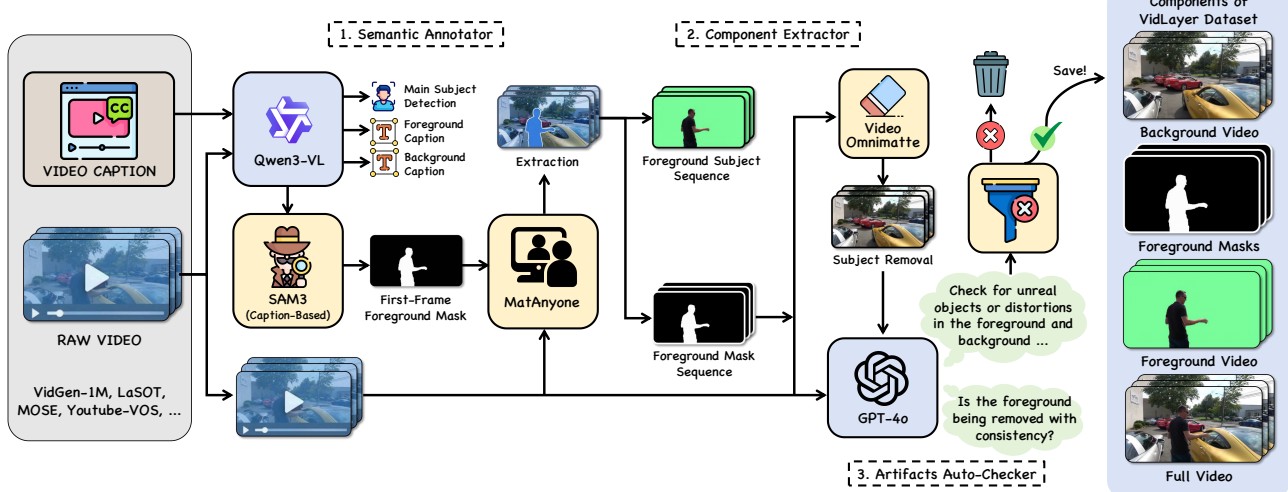

*Figure 2.* Data construction pipeline of *VidLayer* dataset.

may be mixed or misaligned during attention and denoising, harming cross-layer consistency. To address this, we introduce LayerAdaLN and Layered Cross-Attention to explicitly distinguish layer identity.

**Layer Adaptive Normalization (LayerAdaLN).** Different layers exhibit substantially different distributions: sparse near-binary masks, dynamic foregrounds with strong motion, and backgrounds with rich yet largely static textures. To introduce layer-specific modulation while sharing backbone parameters, we maintain a learnable modulation vector for each layer category $l$:

$$\gamma^{(l)} = (b_a^{(l)}, s_a^{(l)}, g_a^{(l)}, b_f^{(l)}, s_f^{(l)}, g_f^{(l)}) \in \mathbb{R}^{6 \times d}, \quad (5)$$

which provides shift, scale, and gate parameters for self-attention and FFN. These vectors are **shared across all blocks** and initialized to zero for stable adaptation.

Given the layer index $l_i$ of token $i$, LayerAdaLN fuses layer modulation with timestep modulation $e_t$ via addition: $e_i = e_t + \gamma^{(l_i)}$. Adaptive normalization is then applied as:

$$\hat{x} = \mathrm{LN}(x) \cdot (1 + s) + b, \quad x \leftarrow x + g \cdot \mathcal{F}(\hat{x}), \quad (6)$$

where $\mathcal{F}$ denotes self-attention or FFN. This design adapts the shared backbone to layer-specific statistics with negligible overhead.

**Layered Cross-Attention Modulation.** To achieve fine-grained semantic control while mitigating condition leakage, we modulate the cross-attention mechanism. We use T5 to independently encode the full-video, foreground, and background prompts, obtaining context embeddings $c_{\mathrm{full}}$, $c_{\mathrm{fg}}$, and $c_{\mathrm{bg}}$, which are concatenated along the sequence dimension into a unified context $c$. This independent encoding prevents

semantic leakage across prompts by construction. We construct an attention mask $M \in \mathbb{R}^{L_{\mathrm{vis}} \times L_{\mathrm{text}}}$ to control visibility between visual and text tokens, where $M_{ij} = 0$ if text token $j$ is visible to visual token $i$, and $-\infty$ otherwise. This mask enforces layer-wise semantic routing: foreground tokens attend only to $c_{\mathrm{fg}}$, background to $c_{\mathrm{bg}}$, full-video to all three embeddings, and mask tokens follow the foreground pattern. The mask is injected as an additive bias:

$$\mathrm{Attention}(Q, K, V) = \mathrm{softmax}\left(\frac{QK^\top}{\sqrt{d}} + M\right)V, \quad (7)$$

where $Q$ is computed from visual tokens, $K$ and $V$ from the text context $c$, and $d$ denotes the attention head dimension.

### 3.3. Training and Inference

LayerT2V adopts a three-stage training strategy to progressively *(i)* adapt the VAE to alpha masks, *(ii)* learn joint multi-layer generation, and *(iii)* multi-foreground extension.

**Stage 1: Mask VAE Adaptation.** We adapt the Wan VAE to support mask encoding and decoding by freezing the encoder $E(\cdot)$ and fine-tuning the decoder $D(\cdot)$ with LoRA, while inserting a lightweight projection head after the encoder to adjust the latent mapping. Freezing the encoder preserves the pretrained latent space distribution, ensuring compatibility with the diffusion backbone in Stage 2. Given a ground-truth alpha mask $A \in [0, 1]$, we replicate it along the channel dimension to match the RGB input format and optimize:

$$\mathcal{L}_{\mathrm{stage1}} = \mathcal{L}_{\mathrm{rec}} + \lambda_{\mathrm{edge}}\mathcal{L}_{\mathrm{edge}} + \lambda_{\mathrm{perc}}\mathcal{L}_{\mathrm{perc}}, \quad (8)$$

where $\mathcal{L}_{\mathrm{rec}}$ is a SmoothL1 reconstruction loss, $\mathcal{L}_{\mathrm{edge}}$ enforces boundary sharpness, and $\mathcal{L}_{\mathrm{perc}}$ encourages structural fidelity.

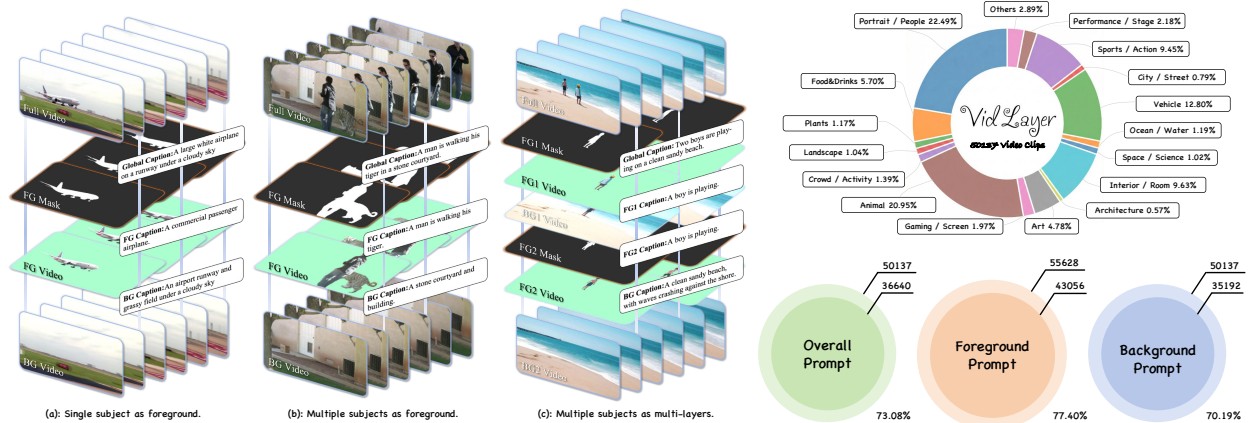

*Figure 3.* **Left**: Visualization samples of VidLayer dataset, it involves layered contents and corresponding layered prompts. **Right**: Scene classifications of VidLayer and semantic redundancy of dataset prompts. For semantic redundancy in text prompts, we extract text embeddings using CLIP (Radford et al., 2021) and set a cosine similarity threshold of 0.85 to identify duplicates.

**Stage 2: Multi-layer Generation Training.** We freeze all VAE components and the text encoder, fine-tune the generation backbone, and train the proposed layer-aware modules. Given a multi-layer training sample, we encode each layer and concatenate the latents along the temporal dimension to form the target latent $z_0 \in \mathbb{R}^{C' \times 4T' \times H' \times W'}$. We sample a shared Gaussian noise $z_1 \sim \mathcal{N}(0, I)$ with the same shape as $z_0$ and construct the linear path $z_t = (1 - t)z_0 + tz_1$, where $t \in (0, 1)$ is sampled from a logit-normal distribution. The basic objective is the Flow Matching loss:

$$\mathcal{L}_{\text{FM}} = \mathbb{E}_{t,z_0,z_1}\left[\left\|v_\theta(z_t, t, c) - (z_1 - z_0)\right\|_2^2\right]. \quad (9)$$

We introduce two auxiliary losses. First, a compositing consistency loss to enforce inter-layer coherence: we recover $\hat{z}_0 = z_t - t\, v_\theta(z_t, t, c)$ and enforce that composed layers match the full video:

$$\mathcal{L}_{\text{cons}} = \left\|\hat{z}_{\text{fg}} + \hat{z}_{\text{bg}} \odot (1 - \tilde{A}) - z_{\text{full}}\right\|_2^2, \quad (10)$$

where $\tilde{A}$ is the downsampled ground-truth mask. This loss provides a strong supervisory signal for layer consistency. Second, to better capture the sparse and near-binary nature of masks, we introduce the mask reconstruction loss:

$$\mathcal{L}_{\text{mask}} = \mathcal{L}_{\text{rec}}(\hat{A}, A) + \lambda_\nabla \mathcal{L}_{\text{grad}}(\hat{A}, A), \quad (11)$$

where $\hat{A}$ is decoded from the predicted mask latent, and $\mathcal{L}_{\text{grad}}$ is a gradient consistency loss that enhances boundary sharpness. The final objective is:

$$\mathcal{L}_{\text{stage2}} = \mathcal{L}_{\text{FM}} + \lambda_{\text{cons}} \mathcal{L}_{\text{cons}} + \lambda_{\text{mask}} \mathcal{L}_{\text{mask}}. \quad (12)$$

**Stage 3: Multi-foreground Extension.** We extend LayerT2V to support multiple foreground subjects by serializing additional foreground-mask pairs along the temporal dimension. The LayerAdaLN module is extended accordingly,

with new foreground layer parameters initialized from the pretrained single-foreground model to ensure stable adaptation. We continue training on the multi-subject subset of VidLayer for 5K steps, enabling LayerT2V to generate two independent foreground layers with their corresponding alpha mattes in a single inference pass.

**Inference.** Given layer-specific text prompts, we first sample a shared Gaussian noise and then perform iterative integration along the ODE trajectory through multiple forward passes. The resulting latent is split along the temporal dimension into individual layer segments, which are decoded by the RGB VAE and Mask VAE respectively to produce the full video, background, foreground(s), and corresponding alpha matte(s).

## 4. VidLayer Dataset

Recent progress in T2V generation has largely focused on synthesizing visually coherent full-frame videos. However, most existing benchmarks and training datasets only provide monolithic video representations, without explicitly modeling the internal compositional structure of a scene. To support our proposed LayerT2V model, we introduce **VidLayer**, a large-scale dataset specifically designed for layer-aware video generation. VidLayer provides aligned foreground videos, foreground masks, background videos, and raw videos, enabling supervision at both the semantic and structural levels. Since such data is not available in the community, constructing VidLayer is a foundational step toward layer-wise text-to-video modeling.

### 4.1. Data Construction Pipeline

As illustrated in Figure 2, VidLayer is constructed through an automated data engine pipeline consisting of three stages: *(i) semantic annotation*, *(ii) multi-layer component extraction*, and *(iii) automatic quality review*. We start from

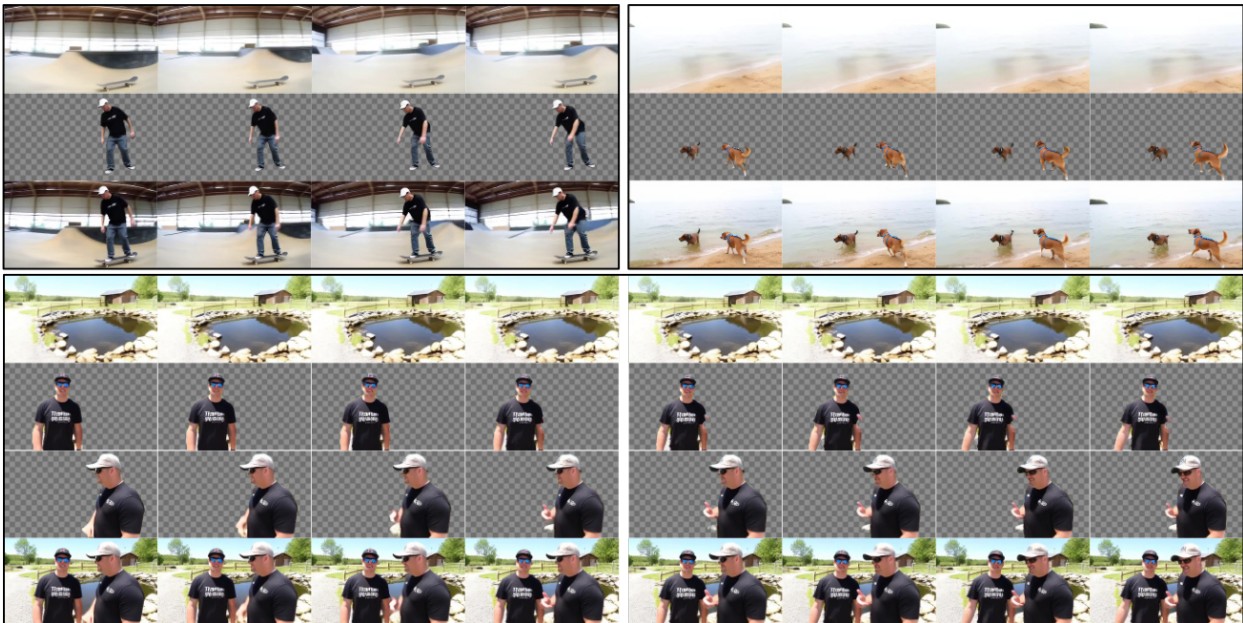

*Figure 4.* **Qualitative results.** LayerT2V generates high-fidelity multi-layer videos across three generation modes: (a) single-foreground with a single subject, (b) single-foreground with multiple subjects, and (c) multi-foreground joint generation with independent layers. Our method produces clean foreground separation, sharp alpha mattes, and complete backgrounds without leakage or boundary artifacts across diverse scenes and motion patterns.

raw caption–video pairs sampled from multiple community datasets and progressively transform them into structured multi-layer video representations. The entire pipeline is fully automated and scalable, allowing us to curate high-quality data at scale without manual intervention.

### 4.2. Semantic Annotation

We first collect over 200k videos from the VidGen (Tan et al., 2024), LaSOT (Fan et al., 2019), MOSE (Ding et al., 2023), Youtube-VOS (Xu et al., 2018), GOT-10k (Huang et al., 2019) datasets, each paired with a natural language caption. Using the video understanding capability of Qwen3-VL (Bai et al., 2025), we perform *subject-oriented semantic annotation* for each video. Specifically, the annotator identifies the primary foreground subject (*e.g*, a person or a salient object) and generates two complementary textual descriptions: a *foreground caption* describing the subject itself, and a *background caption* characterizing the remaining scene context.

To extract foreground subjects, we further exploit the caption-to-mask capability of SAM3 (Carion et al., 2025). By combining the foreground caption with the first video frame, SAM3 produces an initial foreground mask. This mask serves as a reliable spatial prior for subsequent temporally consistent video segmentation.

### 4.3. Multi-layer Component Extraction

Based on the first-frame foreground mask, we employ MatAnyone (Yang et al., 2025) to extract temporally con-

sistent multi-layer components. Given the original video and the initial mask, MatAnyone outputs a sequence of *foreground masks* as well as a corresponding *foreground subject video*, where the subject is rendered on a green-screen background. This process robustly handles challenging scenarios such as occlusion, pose variation, and scale changes, ensuring temporal coherence of the extracted subject. We then apply Gen-Omnimatte (Lee et al., 2025), a video matting method for *foreground removal*. The resulting background preserves realistic spatial structures and temporal continuity without introducing noticeable artifacts.

At the end of this stage, each video is decomposed into a structured set of aligned components: a foreground video, a foreground mask sequence, a background video, and the original full video. We provide additional examples in Figure 19 for enhanced illustration.

### 4.4. Automatic Quality Control

Although the pipeline is fully automated, naive application of video decomposition models can introduce various failure cases. To ensure the reliability and usability of VidLayer, we employ GPT-4o (OpenAI et al., 2024) as an *Artifacts Auto-Checker* to perform strict quality control on the generated components. GPT-4o evaluates multiple criteria, including whether the foreground subject is clear and recognizable, and whether the background videos involve the presence of unrealistic objects, structural distortions, or incomplete human limbs caused by inpainting artifacts. Only samples

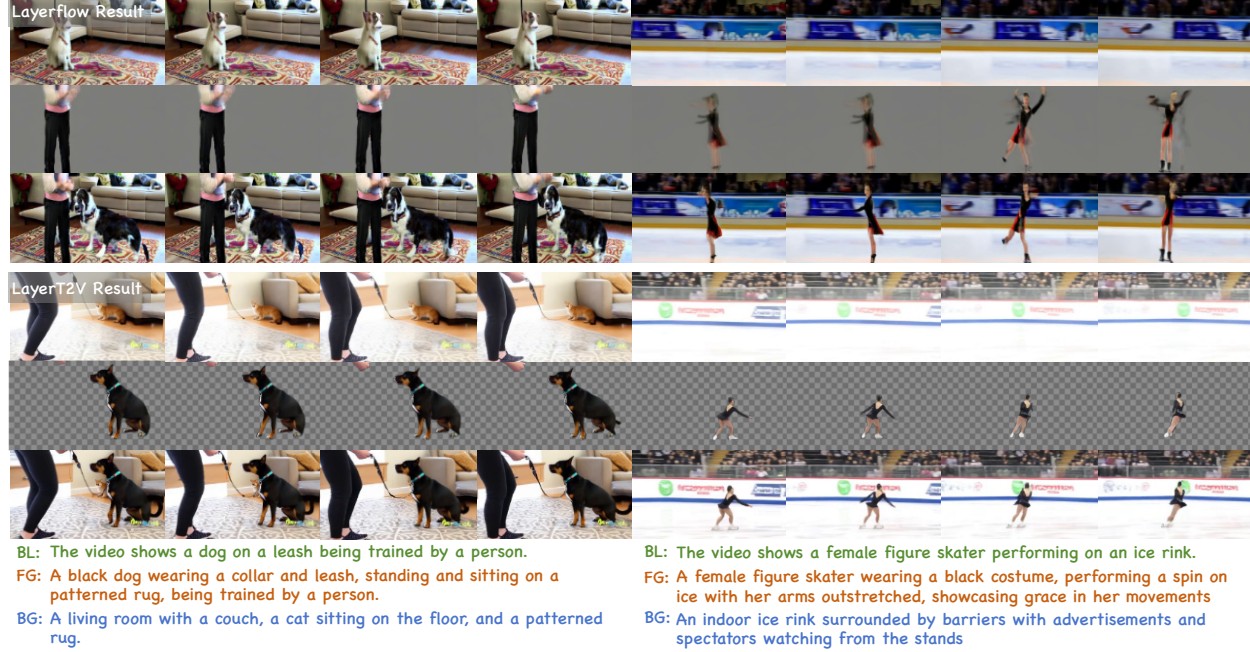

*Figure 5.* **Qualitative comparison.** Compared to LayerFlow, LayerT2V produces higher-quality video layers with stronger temporal consistency and better text alignment. BL/BG/FG correspond to the full-video/background/foreground prompts. Note that LayerFlow outputs the foreground as RGB (without alpha), as it claims RGB foregrounds achieve higher visual quality than RGBA.

that pass all quality checks are retained in the final dataset.

### 4.5. Dataset Statistics and Properties

After automatic filtering, the resulting **VidLayer** dataset contains **50K** high-quality video clips, comprising approximately **4M** frames in total. Among them, around 0.6M frames correspond to multi-subject scenarios involving two or three foreground subjects. Each sample in VidLayer provides aligned multi-layer representations, enabling direct supervision for layer-aware video generation, decomposition, and editing tasks. We also calculate *semantic redundancy* in text prompts and scene classifications to ensure diversity, as shown in Figure 3.

VidLayer fills a critical gap in existing video datasets by offering large-scale, structured multi-layer video data, and serves as a key enabler for LayerT2V model and future research on compositional and controllable video generation.

## 5. Experiments

### 5.1. Experimental Setup

**Implementation Details.** LayerT2V is built upon the pretrained Wan2.1-1.3B (Wan et al., 2025) video generation model. We use the AdamW optimizer with a learning rate of $1 \times 10^{-4}$ and a cosine schedule with 500 warmup steps. All experiments are conducted on 8 NVIDIA H20 GPUs. In Stage 1, we freeze the VAE encoder and fine-tune the decoder with LoRA (rank 128). In Stage 2, we insert LoRA

adapters along with the proposed layer-aware modules into the pretrained backbone. In Stage 3, we continue training on the multi-subject subset of VidLayer to support up to three foreground layers. All stages adopt a progressive training strategy from low to high resolution. Detailed hyperparameters are provided in the appendix A.

**Evaluation Metrics.** We adopt VBench (Huang et al., 2023) as our primary evaluation framework, which provides a comprehensive assessment across multiple dimensions, including subject consistency, temporal flickering, motion smoothness, and aesthetic quality. For text alignment, we use ViCLIP (Wang et al., 2023) to measure the semantic correspondence between generated videos and input prompts. We compare LayerT2V against **LayerFlow** (Ji et al., 2025), a recent multi-layer video generation approach. To ensure fair evaluation, we carefully sample 200 layered prompts from the VidLayer dataset and remove them from our training set. All quantitative results are reported on this fixed held-out set.

### 5.2. Analysis and Discussion

**Qualitative Results.** Fig. 4 visualizes LayerT2V across three generation modes: single-foreground with a single subject, single-foreground with multiple subjects, and multi-foreground joint generation with independent layers. Even under fast motion and complex styles, our method produces clean foregrounds with sharp, temporally stable alpha mattes and complete backgrounds without foreground leakage

*Table 1.* **Quantitative comparison on VBench.** We evaluate five dimensions across foreground (FG), background (BG), and full-video (BL) layers on 200 held-out prompts from VidLayer. Higher values indicate better performance. Best results are highlighted in **bold**.

| Method | Aesthetic Quality ↑ | | | Motion Smoothness ↑ | | | Temporal Flickering ↑ | | | Subject Consistency ↑ | | | Text Alignment ↑ | | |
|---|---|---|---|---|---|---|---|---|---|---|---|---|---|---|---|
| | FG | BG | BL | FG | BG | BL | FG | BG | BL | FG | BG | BL | FG | BG | BL |
| LayerFlow | 0.4591 | 0.4534 | 0.4895 | 0.9582 | 0.9891 | 0.9788 | 0.9630 | 0.9736 | 0.9634 | 0.9440 | 0.9682 | 0.9624 | 0.1727 | 0.1849 | 0.1941 |
| LayerT2V (Native Mask VAE) | 0.4432 | 0.5327 | 0.5010 | 0.9863 | 0.9908 | 0.9776 | 0.9661 | **0.9837** | 0.9669 | 0.9828 | **0.9839** | 0.9749 | 0.1846 | 0.2102 | 0.1996 |
| **LayerT2V (VAE LoRA)** | **0.4971** | **0.5577** | **0.5391** | **0.9919** | **0.9920** | **0.9845** | **0.9872** | 0.9830 | **0.9676** | **0.9829** | 0.9838 | **0.9750** | **0.2009** | **0.2307** | **0.2136** |

*Table 2.* **User study results.** Values indicate the fraction of times each method is selected as the best (higher is better). Best results are highlighted in **bold**.

| Method | Aesthetic ↑ | FG Quality ↑ | Text Align. ↑ |
|---|---|---|---|
| LayerFlow | 0.120 | 0.156 | 0.136 |
| LayerT2V (Native Mask VAE) | 0.156 | 0.076 | 0.196 |
| **LayerT2V (Ours)** | **0.724** | **0.768** | **0.668** |

*Table 3.* **Ablation study on RoPE configuration.** 3D RoPE preserves the pretrained spatiotemporal encoding and consistently outperforms 4D variants that reallocate dimensions to a layer axis.

| Configuration | Subject Consistency↑ | | | Temporal Flickering↑ | | | Text Alignment↑ | | |
|---|---|---|---|---|---|---|---|---|---|
| | FG | BG | BL | FG | BG | BL | FG | BG | BL |
| 4D RoPE (38,38,40,12) | 0.964 | 0.970 | 0.958 | 0.952 | 0.965 | 0.949 | 0.195 | 0.224 | 0.209 |
| 4D RoPE (40,40,42,6) | 0.971 | 0.976 | 0.965 | 0.966 | 0.973 | 0.957 | 0.198 | 0.227 | 0.211 |
| **3D RoPE (42,42,44)** | **0.983** | **0.984** | **0.975** | **0.987** | **0.983** | **0.968** | **0.201** | **0.231** | **0.214** |

or boundary flickering. Fig. 5 presents the qualitative comparison results: LayerT2V shows consistently higher-quality layers and more consistent recompositions.

**Quantitative Evaluation.** We adopt VBench (Huang et al., 2023) to evaluate foreground (FG), background (BG), and blending results (BL) across five dimensions: *Aesthetic Quality*, *Motion Smoothness*, *Temporal Flickering*, *Subject Consistency*, and *Text Alignment*. As shown in Table 1, LayerT2V achieves strong performance across all metrics. The FG exhibits high subject consistency with minimal boundary jitter, while the BG remains temporally coherent without foreground leakage. The recomposited BL preserves favorable scores, validating both individual layer quality and cross-layer coherence.

**User Study.** We also conduct a user study to evaluate the perceptual quality of generated multi-layer videos. Participants are asked to compare results from LayerFlow, LayerT2V with Native Mask VAE, and our LayerT2V across three aspects: Aesthetic Quality, Foreground Quality and Text Alignment. As shown in Table 2, LayerT2V is consistently preferred over all baselines, demonstrating superior layer separation and visual quality.

### 5.3. Ablation Study.

**4D RoPE for Layer-aware Position Embedding.** A natural idea is to extend the standard 3D RoPE (Su et al., 2021),

*Table 4.* **Ablation study on layer-aware modules.** Baseline: vanilla temporal concatenation only. ①: LayerAdaLN, ②: Layer-CrossAttention. Best results are highlighted in **bold**.

| Configuration | Subject Consistency↑ | | | Temporal Flickering↑ | | | Text Alignment↑ | | |
|---|---|---|---|---|---|---|---|---|---|
| | FG | BG | BL | FG | BG | BL | FG | BG | BL |
| Baseline | 0.931 | 0.942 | 0.924 | 0.955 | 0.961 | 0.945 | 0.169 | 0.182 | 0.188 |
| Baseline + ① | 0.973 | 0.975 | 0.961 | 0.979 | 0.976 | 0.959 | 0.180 | 0.194 | 0.195 |
| Baseline + ② | 0.941 | 0.953 | 0.938 | 0.964 | 0.969 | 0.953 | 0.191 | 0.218 | 0.206 |
| **Baseline + ① + ②** | **0.983** | **0.984** | **0.975** | **0.987** | **0.983** | **0.968** | **0.201** | **0.231** | **0.214** |

which operates over the temporal and spatial dimensions, to a 4D variant by introducing an additional layer axis. However, as shown in Fig. 6 (a) and (b), this proves ineffective: rather than promoting disentanglement, 4D RoPE interferes with the pretrained positional embedding, causing interframe flickering and boundary artifacts. We attribute this to the fact that layer identity is a categorical attribute rather than a spatial or temporal dimension—positional embedding alone cannot disentangle the artificial layer ordering from natural temporal dynamics, necessitating dedicated layer conditioning mechanisms.

This observation is quantitatively confirmed in Table 3. Both 4D variants underperform the 3D baseline across all metrics, and the degradation grows as more capacity is diverted to the layer axis. The standard 3D configuration $(42, 42, 44)$, which preserves the full pretrained spatiotemporal encoding, attains the best Subject Consistency, Temporal Flickering, and Text Alignment on all three layers. This indicates that the benefit of an explicit layer axis is outweighed by the disruption it causes to the pretrained positional structure, reinforcing our choice to handle layer identity through dedicated conditioning rather than position embedding.

**Alpha Mask Processing.** We compare two strategies: training a native Mask VAE from scratch versus our VAE LoRA method. As shown in Fig. 6 (c) and (d), the native Mask VAE produces discernible layer decomposition, validating that treating alpha masks distinctly from RGB is feasible. However, it exhibits significant quality deficiencies: temporal flickering, blurry boundaries, and inconsistent alpha values, resulting in poor mask fidelity. We attribute these artifacts to the lack of pretrained video priors. In contrast, VAE LoRA leverages the spatial and temporal representations of the pretrained Wan VAE, applying min-

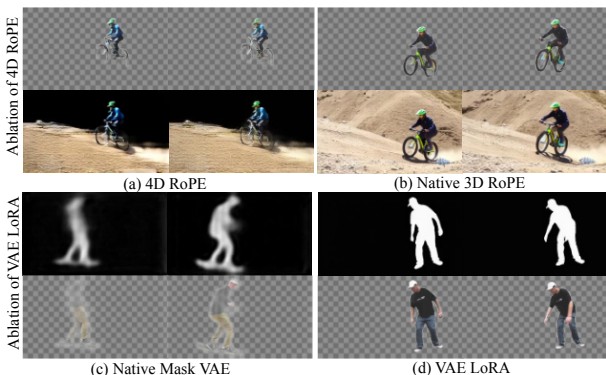

**Figure 6. Ablation results.** (a-b) Using 4D RoPE disrupts the pretrained spatiotemporal positional encoding, severely degrading generation quality. (c-d) Compared to the Native Mask VAE, our VAE LoRA strategy produces higher-quality foreground masks.

*Table 5.* **Ablation study on GPT-4o auto-check for data filtering.** Models are trained under matched data scale and identical settings on the unfiltered versus the GPT-4o-filtered dataset. Best results are highlighted in **bold**.

| Configuration | Subject Consistency↑ | | | Temporal Flickering↑ | | | Text Alignment↑ | | |
|---|---|---|---|---|---|---|---|---|---|
| | FG | BG | BL | FG | BG | BL | FG | BG | BL |
| w/o auto-check | 0.959 | 0.968 | 0.972 | 0.955 | 0.967 | 0.961 | 0.172 | 0.219 | 0.211 |
| **w/ auto-check** | **0.983** | **0.984** | **0.975** | **0.987** | **0.983** | **0.968** | **0.201** | **0.231** | **0.214** |

imal trainable parameters to extend the decoder for mask reconstruction while preserving RGB encoding quality.

**Layer-Aware Modules.** Table 4 validates that both Layer-AdaLN and Layered Cross-Attention are essential. The baseline suffers from foreground-background entanglement, low subject consistency and poor text alignment due to semantic leakage. Adding LayerAdaLN improves layer separation and temporal stability, yet text alignment remains suboptimal as cross-attention still allows inter-layer contamination. The full model achieves the best results: LayerAdaLN ensures clean decomposition, while Layered Cross-Attention enforces strict visibility constraints, maximizing consistency and alignment across all layers.

**Ablation of the GPT-4o Auto-check.** To validate the effectiveness of the GPT-4o auto-check, we compare models trained on the original unfiltered dataset and the GPT-4o-filtered dataset under matched data scale and identical settings. As shown in Table 5, training on filtered data yields consistent gains across Subject Consistency, Temporal Flickering, and Text Alignment. This indicates that filtering out samples with incorrect layer decomposition leads to cleaner and more stable generation, confirming the effectiveness of the GPT-4o auto-check.

# 6. Conclusion

We present **LayerT2V**, a unified multi-layer video generation framework that produces semantically consistent full videos, background layer, and multiple foreground RGB layers with corresponding alpha mattes in a single inference pass. To address layer-identity ambiguity and conditional leakage, we design LayerAdaLN and layer-aware cross attention modulation. We also release **VidLayer**, the first large-scale multi-layer video dataset. Experiments demonstrate that LayerT2V substantially outperforms prior methods in visual fidelity, temporal consistency, and cross-layer coherence. We hope this work advances multi-layer video generation and enables fine-grained control in professional video production.

# Acknowledgements

We would like to greatly thank Ming-Hsuan Yang and Kelvin C.K. Chan for their insightful discussions and generous support. We also thank Yangnan Lin for his help in testing benchmarks for our model. This work was supported by the National Natural Science Foundation of China under Grant 62572317.

# Impact Statement

**Positive Societal Impact.** Our work democratizes access to advanced video production techniques that were previously limited to professionals with specialized software and expertise. By providing editable layered representations, LayerT2V can accelerate content creation workflows in filmmaking, advertising, education, and entertainment. The release of the VidLayer dataset and our code will further support research in compositional video generation and facilitate reproducibility.

**Potential Risks and Mitigation.** As with other generative video technologies, LayerT2V could potentially be misused for creating misleading or deceptive content. The ability to seamlessly replace backgrounds or modify foreground subjects may raise concerns about authenticity and misinformation. We encourage the research community and practitioners to develop and adopt detection mechanisms, watermarking techniques, and content provenance standards to mitigate these risks. We also advocate for responsible use guidelines and transparency in disclosing AI-generated content.

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

## A. Implementation Details

LayerT2V is built upon the pretrained Wan2.1-1.3B (Wan et al., 2025) video generation model. We use the AdamW optimizer with a learning rate of $1 \times 10^{-4}$, weight decay of 0.01, and a cosine learning rate schedule with 500 warmup steps. Timesteps are sampled from a logit-normal distribution with mean 0.5 and standard deviation 1.0. All experiments are conducted on 8 NVIDIA H20 GPUs.

*Stage 1: Mask VAE Adaptation.* We freeze the Wan VAE encoder and fine-tune the decoder with LoRA (rank 128). The loss weights are set to $\lambda_{\mathrm{edge}} = 1.0$ and $\lambda_{\mathrm{perc}} = 1.0$. We adopt a progressive training strategy: first training at $192 \times 336$ resolution with 9 frames (batch size 8, 4.5K steps), then scaling to $384 \times 672$ with 41 frames (batch size 2, 2.5K steps).

*Stage 2: Multi-layer Generation.* Starting from the pretrained Wan2.1 backbone, we insert LoRA adapters (rank 128) along with the proposed LayerAdaLN and layer-aware cross-attention modules. Loss weights are set to $\lambda_{\mathrm{cons}} = 0.1$, $\lambda_{\mathrm{mask}} = 0.1$, and $\lambda_{\nabla} = 0.1$. We first train at $192 \times 336$ with 9 frames (batch size 8, 6K steps), then fine-tune at $384 \times 672$ with 41 frames (batch size 4, 1K steps).

*Stage 3: Multi-foreground Extension.* Using the same training configuration as the high-resolution phase of Stage 2 ($384 \times 672$, 41 frames, batch size 1), we continue training for 5K steps. This stage extends the model to support up to three independent foreground layers with their corresponding alpha mattes.

## B. Computational Cost

We analyze the inference efficiency of LayerT2V on a single NVIDIA H100 GPU. Since LayerT2V jointly generates four spatially aligned outputs (full video, background, foreground, and alpha matte) in a single forward pass, wall-clock runtime alone does not fully reflect its effective generation throughput. We therefore additionally report *Effective Output FPS*, which counts all generated frames across the aligned outputs and normalizes by runtime.

As summarized in Table 6, LayerT2V generates four aligned layers with 164 total frames at $672 \times 384$ resolution, requiring 24.4 GB of memory and a runtime of 157 s. For reference, the original Wan 2.1 1.3B backbone does not support $672 \times 384$; at $832 \times 480$ it generates a single 41-frame video using 18.1 GB and 40 s. These results indicate that, although LayerT2V has a longer wall-clock runtime, its effective throughput is comparable to the single-layer baseline once normalized by the total number of generated outputs (1.03 FPS for both).

*Table 6.* **Inference cost of LayerT2V.** Measured on a single NVIDIA H100 GPU. Effective Output FPS counts all generated frames across the aligned outputs, normalized by runtime, to reflect the throughput of joint multi-layer generation.

| Method | Resolution | Output | Peak Mem. | Runtime (s) | Eff. Output FPS |
|---|---|---|---|---|---|
| LayerT2V | $672 \times 384$ | 4 aligned layers, 164 frames | 24.4 GB | 157 | 1.03 |
| Wan 2.1 1.3B | $832 \times 480$ | 1 video, 41 frames | 18.1 GB | 40 | 1.03 |

## C. Analysis of Layer-aware Modules

Our main paper establishes the quantitative benefit of LayerAdaLN and LayeredCrossAttention through ablation. Here we provide further analysis that explains *why* these two modules are effective, based on the learned modulation parameters and the resulting attention behavior.

**LayerAdaLN.** When the four layers (full/blending video, background, foreground, and alpha matte) are serialized and processed by a shared DiT backbone, they exhibit different statistical characteristics, and a single set of modulation parameters forces them to share one normalization solution. LayerAdaLN instead assigns each layer its own shift/scale/gate parameters and fuses them with the timestep modulation at every block, reducing inter-layer entanglement while preserving the pretrained video prior. To verify that the network actually exploits this added capacity, we inspect the learned modulation vectors. Figure 7 reports the pairwise cosine similarity between the four layers' modulation vectors for each shift/scale/gate component in self-attention and FFN. The off-diagonal similarities are consistently small in magnitude—for the shift components they are even weakly negative (around $-0.36$ in attention and $-0.46$ in FFN between the full-video and background layers)—showing that different layers occupy distinct directions in modulation space rather than collapsing

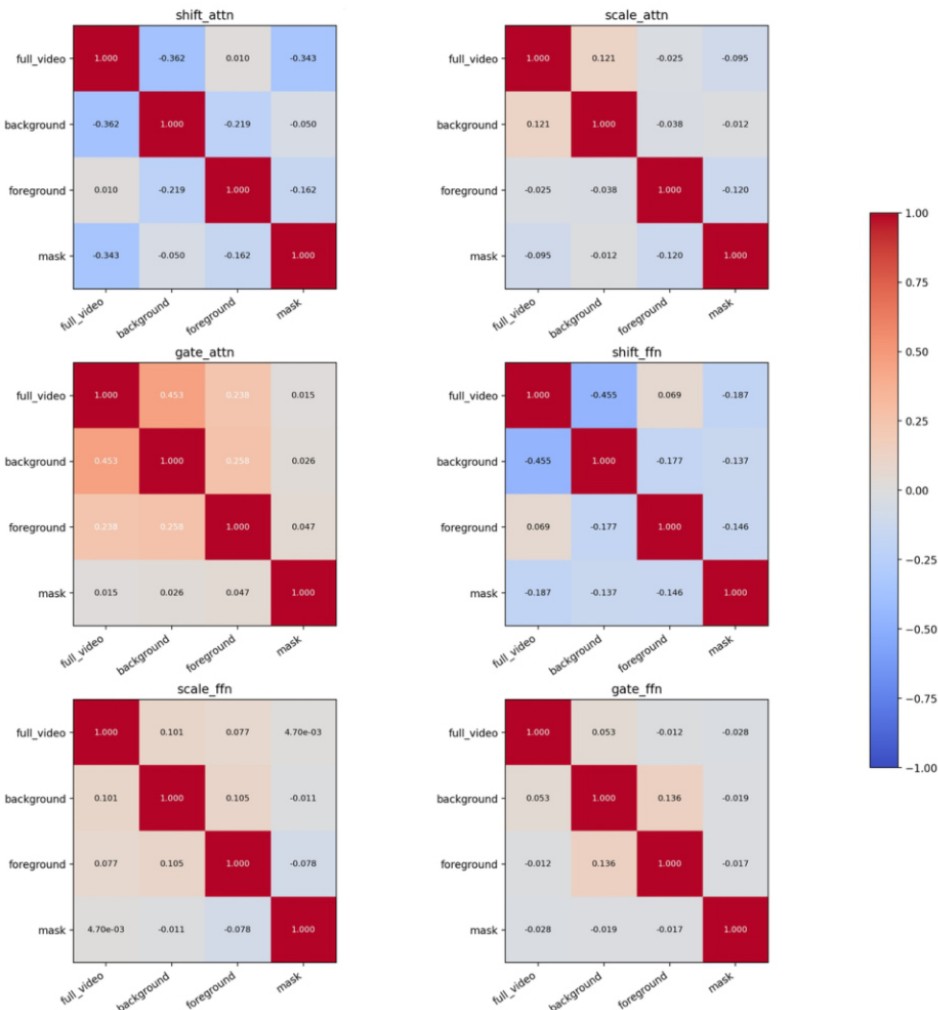

*Figure 7.* **Pairwise cosine similarity of learned LayerAdaLN modulation across layers.** Computed for the blending (full) video, background, foreground, and alpha-mask branches, separately for the shift/scale/gate components in self-attention and FFN. Different layers occupy distinct directions rather than collapsing to a shared solution.

to a shared solution. Figure 8 further shows that the $L_2$ norms of these vectors differ systematically across layers and components: the gate components are consistently smaller than shift/scale, and the alpha-mask branch tends to have the largest magnitude in most components. Together these observations indicate that LayerAdaLN learns both layer-specific *directions* and layer-specific *strengths* of modulation, consistent with its ablation gains in subject consistency and temporal flickering and its modest improvement in text alignment.

**LayeredCrossAttention.** Text conditioning is the main source of cross-layer leakage: when all layers attend to a shared pool of prompts, the blending prompt and other layers' prompts bleed into each layer, producing artifacts and conditional leaks. LayeredCrossAttention suppresses this leakage directly at the cross-attention logit level by restricting each layer to attend only to its own prompt, following the visibility relationship between layers. Figure 9 visualizes the resulting cross-attention maps, averaged over heads and time in the eighth block. With vanilla cross-attention (Figure 9 (a)), the attention is dense and off-target: every image layer attends strongly to the blending prompt (scores of 0.86–0.89), so foreground and background are partly driven by prompts that do not belong to them. With LayeredCrossAttention enabled (Figure 9 (b)), the attention becomes block-sparse and aligned with the intended routing—the blending layer attends to the blending prompt, the background layer to the background prompt, and the foreground and alpha-matte layers to the foreground prompt—while all cross-layer entries drop to zero. This confined routing explains why LayeredCrossAttention

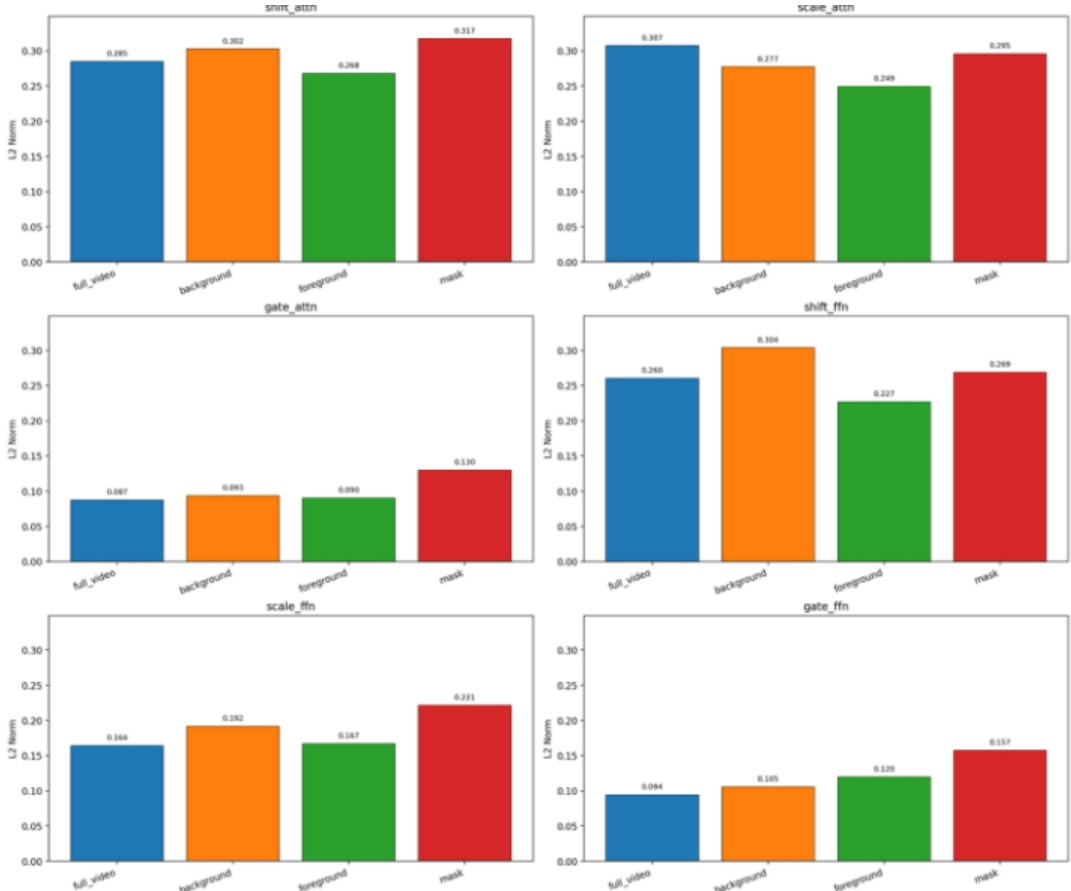

*Figure 8.* $L_2$ **norms of learned LayerAdaLN modulation across layers.** The modulation magnitude varies systematically across branches and components, indicating that LayerAdaLN learns not only different directions but also different strengths of modulation.

improves text alignment and reduces artifacts in the ablation.

## D. Data Construction Details

### D.1. Semantic Annotation and Components Extraction

We use Qwen3-VL-30B-A3B-Instruct (Bai et al., 2025) to detect main foreground subject and obtain layer-aware prompts. The prompt we use for this stage is visualized in the following prompt box:

---

**System Message for Qwen3-VL-30B-A3B-Instruct**

```
You are a video content analysis assistant.

Analyze the video and output STRICT JSON ONLY.

Tasks:
1. Find 1 to 2 main subjects
2. Determine if the main subject is always visible
3. Estimate subject size category
4. Determine camera stability
5. Rate overall video quality

Rules for subjects:
- Subject must be a single object (not a group)
- Subject occupies at least 10% of the frame
- Subject occupies at most 70% of the frame
- If no valid subject exists, output empty subjects list
```

---

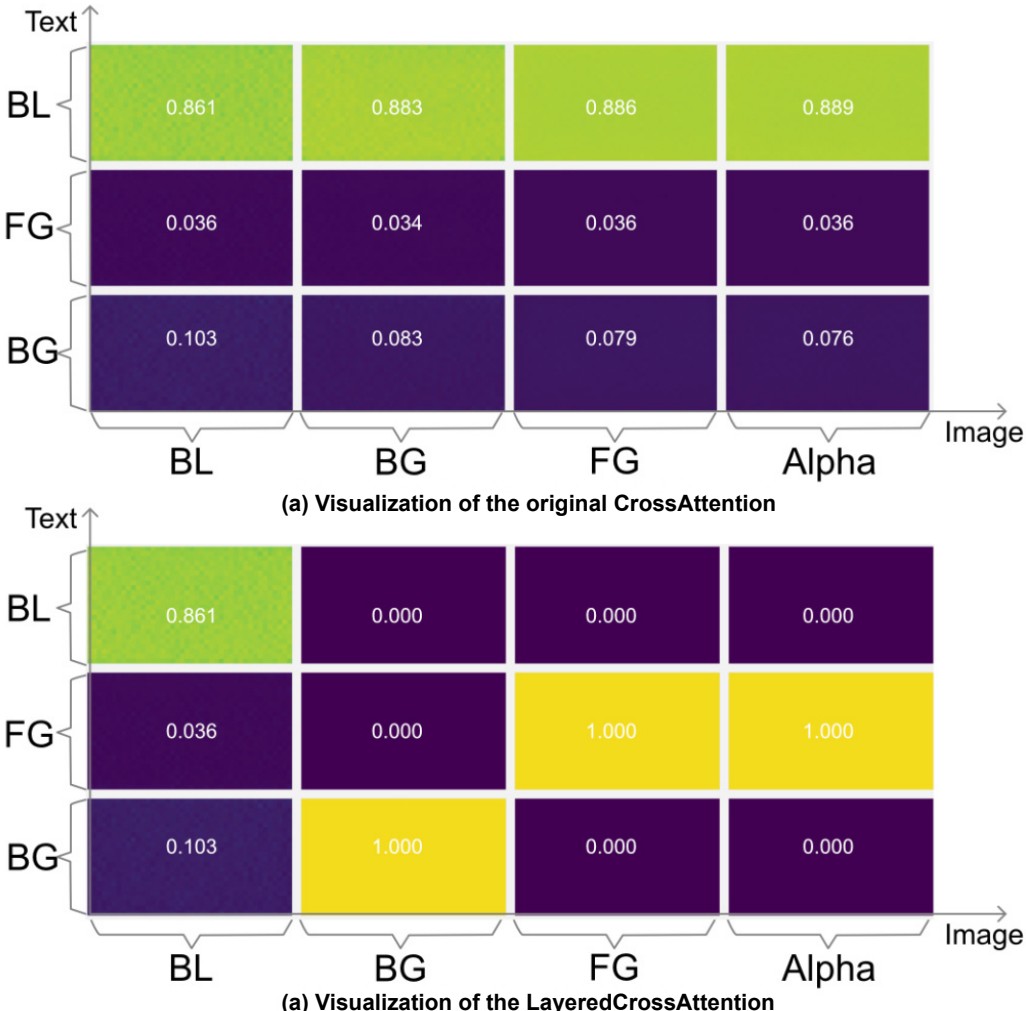

**(a) Visualization of the original CrossAttention**

**(a) Visualization of the LayeredCrossAttention**

*Figure 9.* **Cross-attention map visualization.** Attention scores of the eighth block, averaged over heads and time. Without Layered Cross-Attention (a), layers interact densely with off-target prompts, causing artifacts and conditional leaks. With Layered Cross-Attention (b), each layer is confined to its own prompt.

```
Definitions:
- subject_size must be one of: "small", "medium", "large"
- video_quality is an integer from 0 to 10

Output format:
{
  "subjects": [
    {
      "subject_id": "<string>",
      "subject_description": "<string>"
    }
  ],
  "subject_visibility": {
    "always_visible": true,
    "subject_size": "medium"
  },
  "camera_stability": {
    "stable": true
  },
  "video_quality": 7
}
```

Then prompt-to-mask capability of SAM3 is utilized to extract first-frame mask of the video foreground, as visualized in Figure 10, as a reliable cornerstone for the following sequential masks extraction by MatAnyone (Yang et al., 2025).

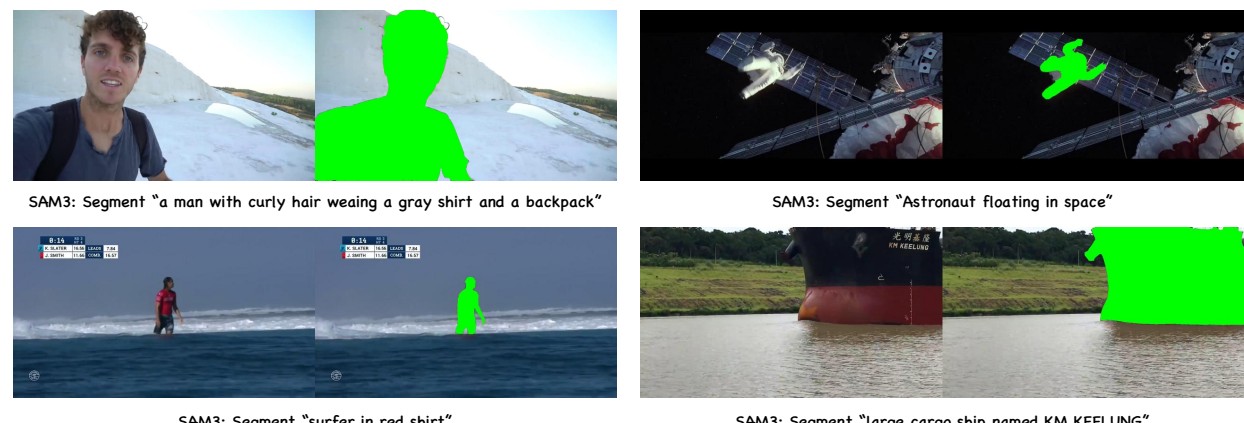

SAM3: Segment "a man with curly hair weaing a gray shirt and a backpack"

SAM3: Segment "Astronaut floating in space"

SAM3: Segment "surfer in red shirt"

SAM3: Segment "large cargo ship named KM KEELUNG"

*Figure 10.* SAM3 (Carion et al., 2025) takes foreground caption as input, and output its local region with binary mask.

### D.2. Details for Artifacts Auto Check

To ensure the visual fidelity and decomposition quality of the VidLayer dataset, we implement an automated quality assurance pipeline leveraging the multimodal capabilities of GPT-4o (OpenAI et al., 2024). For each sampled key-frame, we construct **a composite visual prompt** by horizontally concatenating the original video frame, the object mask, the inpainted background, and the extracted foreground, enabling the model to perform direct reference-based comparisons, as demonstrated in Figure 11. We employ a rigorous scoring rubric that strictly penalizes hallucinations—such as "ghosting" artifacts, color leakage, and geometric distortions—while distinguishing plausible scene interactions (*e.g.*, cast shadows) from unwanted residues, quantifying quality via dual metrics for background reconstruction and foreground extraction.

Ultimately, temporal segments are validated based on the consistency of high scores across adjacent key-frames, where only sequences demonstrating superior layer separation are cropped and incorporated into the final dataset. For example, we take intervals between key-frames as $\Delta_{key} = 10$, then if both *BG Score* and *FG Score* of frame $N$ and frame $N + \Delta_{key}$ are $\geq 7$, we consider frame interval $[N, N + \Delta_{key}]$ as qualified. The system prompt we use for this stage is given in the following prompt box:

---

**System Message for GPT-4o**

```
You are a Data Quality Expert for a Layered Video Generation Model.
Your goal is to curate a training dataset where a video is decomposed into:
1. [BG] Background Layer.
2. [FG] Foreground Layer.

Context Description: {description}

### INPUT LAYOUT
Each image consists of **4 horizontally concatenated panels** (Left to Right):
- **Panel 1 [Original]**: The ground truth video frame.
- **Panel 2 [Mask]**: White area indicates the foreground object location.
- **Panel 3 [BG]**: The inpainted background result (The object should be gone).
- **Panel 4 [FG]**: The extracted foreground result (The object on black).

### VISUAL COMPARISON STRATEGY
**Step 1: Evaluate Foreground (fg_score)**
- Compare Panel 4 [FG] directly against Panel 1 [Original].
- Check: Sharpness, edge accuracy, and semantic preservation.

**Step 2: Evaluate Background (bg_score) - CRITICAL STEP**
- Look at Panel 2 [Mask] to find the region of interest.
- **COLOR LEAKAGE CHECK**: Look at the color of the person in Panel 1 (e.g., Red shirt, Brown hair). Now look at
    Panel 3 [BG] in that same spot.
  - Is there a blurry blob of the **SAME COLOR** (e.g., a red or brown smudge) floating there?
  - If YES -> This is "Color Leakage/Ghosting", NOT a shadow. Score must be **0-3**.
```

```
- **SHADOW VS. BLOB CHECK**:
    - Real shadows are usually dark/grey and cast on the floor/wall.
    - **Floating, colored, amorphous clouds** are artifacts.

### SCORING RUBRIC
**BG SCORING (First Value)**
- **0-3 (FATAL - REJECT)**:
    * **Ghosting/Color Leakage**: A blurry color blob that matches the foreground object's color.
    * **Floating Blobs**: Large amorphous shapes floating in the middle of the room.
    * **Paridolia**: Human features (eyes, face) appearing in the foreground mask region (outside mask region is
        acceptable).
    * **Melting textures**.
- **4-6 (Mediocre)**: Blurry, smudged, visible clone-stamping, but no scary ghosts.
- **7-9 (Good)**: Clean.
    * **ACCEPTABLE**: Sharp/Real props (strings, balls).
    * **ACCEPTABLE**: **Cast Shadows** (Must be dark/grey and geometrically consistent on surfaces).
- **10 (Perfect)**: Flawless.

**FG SCORING (Second Value)**
- **0-3 (Poor)**: Unrecognizable blob, heavy noise.
- **4-6 (Fair)**: Blurry, rough edges.
- **7-9 (Good)**: Clear, recognized, slight halos.
- **10 (Perfect)**: Crystal clear.

### OUTPUT FORMAT
Return a strictly valid JSON object.
Keys: Frame Index (string).
Values: **A list of exactly two integers: [bg_score, fg_score]**.

Example: {{ "0": [9, 8], "10": [2, 9], "20": [10, 10] }}
```

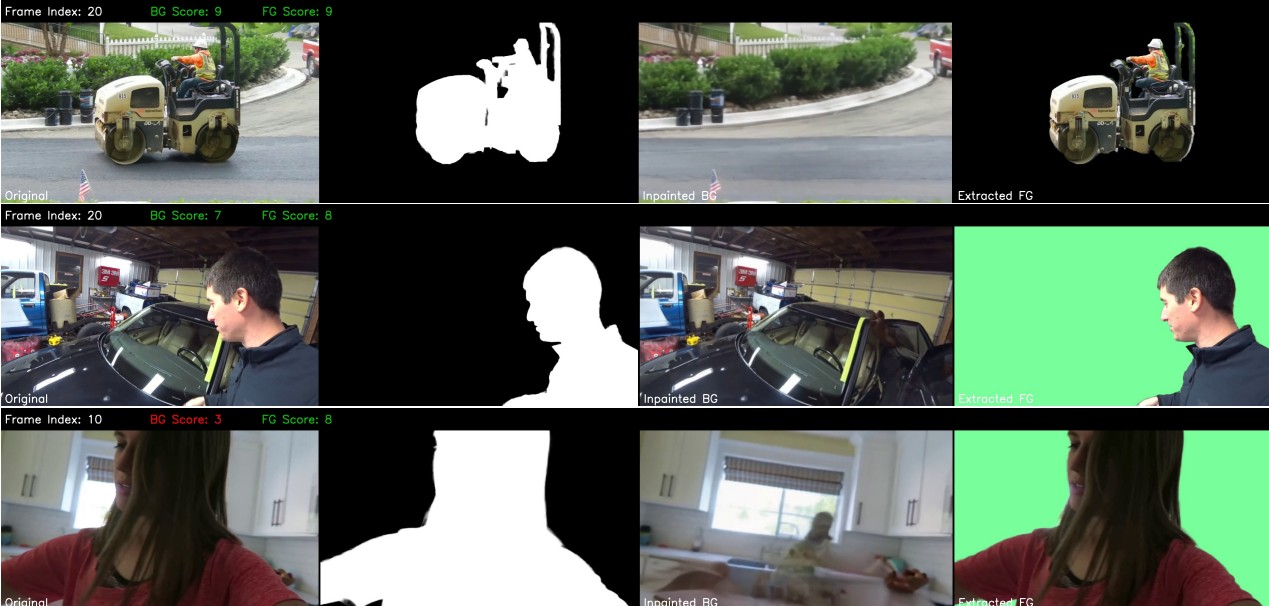

*Figure 11.* GPT-4o quality check results. In the first two rows, the foreground is removed with excellent consistency. But in the third row, there appears some texture leakage from the foreground to the background. This phenomenon can be attributed to the drawbacks of the Gen-Omnimatte (Lee et al., 2025) model. GPT-4o will label such cases as low scores and remove them from the dataset.

## E. Human Validation of the GPT-4o Auto-check

To filter low-quality samples, our pipeline scores each candidate with GPT-4o. Specifically, GPT-4o is presented with a four-panel composite—the original frame, the predicted mask, the inpainted background, and the extracted foreground—and assigns a background (BG) and foreground (FG) quality score on a 0–10 scale, following an explicit rubric that penalizes ghosting, color leakage, floating blobs, and geometric distortions. An interval is kept only if both of its endpoint key frames satisfy $BG \geq 7$ and $FG \geq 7$.

*Table 7.* **Quantitative validation of the GPT-4o auto-check against human judgment.** The most important quantity for dataset curation is the precision of GPT-kept intervals, which reaches 100.0% under human validation.

| Metric | Result |
|---|:---:|
| *Protocol* | |
| Sampled source videos | 100 |
| Sampled candidate intervals | 100 |
| *Interval-level agreement* | |
| GPT-kept intervals also judged keep by humans | **50/50 = 100.0%** |
| GPT-rejected intervals also judged reject by humans | $44/50 = 88.0\%$ |
| Overall agreement | $94/100 = 94.0\%$ |
| Human inter-annotator agreement (Fleiss' $\kappa$) | 0.93 |
| *Score-level agreement* | |
| Spearman $\rho$ between GPT BG score and mean human BG score | 0.96 |
| Spearman $\rho$ between GPT FG score and mean human FG score | 0.90 |

To verify that this automatic rule agrees with human perception, we conduct a human study on 100 candidate intervals sampled from the pre-filter pool (50 GPT-kept and 50 GPT-rejected, one interval per source video), using the same frame gap $\Delta T$ as the generation pipeline. Three annotators, blinded to the GPT-4o outputs and to one another, score the same endpoint composites with the identical 0–10 BG/FG rubric; interval-level labels are then derived using the same endpoint rule and majority vote. For score-level analysis, we compare the GPT-4o scores against the mean human scores using Spearman's rank correlation $\rho$.

The results are summarized in Table 7. GPT-4o aligns closely with human judgment: 100.0% of GPT-kept intervals are also judged keep by humans, 88.0% of GPT-rejected intervals are also judged reject, and the overall interval-level agreement reaches 94.0%. The high human inter-annotator agreement (Fleiss' $\kappa = 0.93$) indicates that the task is well-defined and the human labels are reliable. At the score level, GPT-4o correlates strongly with the mean human scores, with $\rho = 0.96$ for BG and $\rho = 0.90$ for FG. The most important quantity for dataset curation is the precision of GPT-kept intervals, which reaches 100.0% under human validation. Figures 13-14 show qualitative examples covering both kept and rejected intervals, illustrating that GPT-4o reliably identifies decomposition failures such as residual foreground and background artifacts. Together, these results confirm the effectiveness of GPT-4o as an automatic quality filter.

## F. User Study

**Study Design.** We conduct a user study to evaluate perceptual quality aspects that automated metrics may not fully capture. We recruit 30 participants, including both individuals with video editing experience and general users. We randomly sample 25 prompts from the VidLayer test set, covering diverse subjects including humans, animals, and objects. For each prompt, we generate videos using three methods: LayerFlow (Ji et al., 2025), LayerT2V with Native Mask VAE, and LayerT2V with VAE LoRA (Ours).

**Evaluation Protocol.** For each prompt, participants are presented with the three generated videos in randomized order (labeled A, B, C) to eliminate ordering bias. They are asked to evaluate three aspects:

- **Aesthetic Quality**: Overall visual appeal, including color harmony, composition, and absence of artifacts.

- **Foreground Quality**: Completeness, clarity, and temporal stability of the foreground layer and its alpha matte.

- **Text Alignment**: How well the generated video content matches the input text prompt.

For each aspect, participants select the single best video among the three options. The study interface is shown in Fig. 12.

**Results.** We compute the preference rate for each method as the fraction of times it is selected as the best across all participants and prompts. As reported in Table 2 of the main paper, LayerT2V (Ours) achieves the highest preference rates

**1** Which result is the most visually pleasing overall? *

Higher overall visual quality: sharpness, clean details, good lighting/color, fewer artifacts. Better composition and harmony: coherent layout, natural appearance, consistent style. Prefer results that look "complete" and high-fidelity, not noisy or distorted. Answer: Select one: A / B / C.

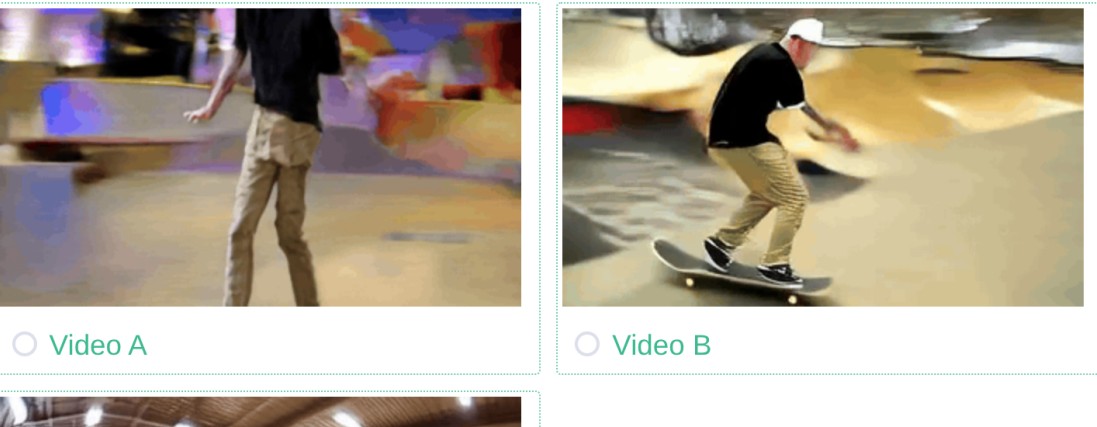

○ Video A          ○ Video B

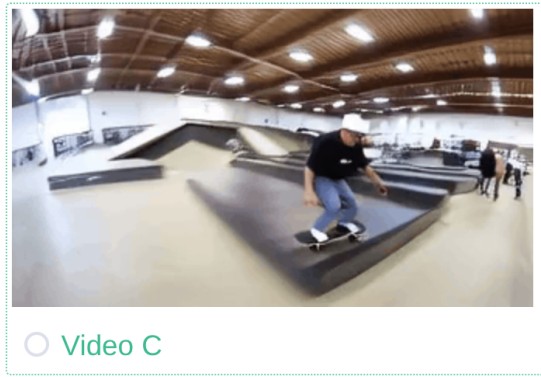

○ Video C

*Figure 12*. **User study interface.** Participants are shown video results from three methods in randomized order and asked to select the best one for each evaluation aspect.

across all three aspects (Aesthetic: 0.724, FG Quality: 0.768, Text Alignment: 0.668), with particularly strong performance on Foreground Quality. This confirms that our method produces visually superior and more coherent multi-layer videos compared to baselines.

## G. More Visual Samples

In this section, we provide additional visual results to complement the main paper. We first present more samples from our VidLayer dataset (Fig. 19). We then include extensive generations of LayerT2V (Figs. 21–23), where each example shows eight sampled frames together with the predicted background, transparent foreground, and recomposited video, as well as the full-video prompt and layer-specific descriptions. These results further demonstrate the visual fidelity, temporal stability, and cross-layer coherence of LayerT2V across diverse motions, appearances, and scenes. Lastly, we present additional demonstrations of LayerT2V with two foreground layers in Figure 24. Owing to computational and resource constraints, our current implementation is limited to modeling two foreground layers. Extending the framework to support a larger number of foreground layers constitutes a promising direction for future work.

## H. Video Editing Applications

A key benefit of LayerT2V is that it directly supports video editing. Since it produces a temporally aligned full video, a background layer, foreground RGB layers, and their alpha masks in a single pass, these structured outputs can be used to edit a scene without regenerating it. This enables two editing modes: background editing, where the background is replaced or modified while the foreground is preserved, and foreground editing, where the background is kept unchanged and only

the target foreground is edited. Figures 15–18 show several editing examples produced with LayerT2V.

## I. Limitations

VidLayer is currently limited to at most three foreground layers, with the majority of samples containing only a single foreground. This limitation stems from two factors: (1) the inherent difficulty of collecting source videos that can be cleanly decomposed into multiple semantically distinct and interacting foreground entities, and (2) the limited robustness of current decomposition models when handling complex scenes.

Specifically, MatAnyone (Yang et al., 2025) struggles to accurately segment small foreground objects over extended temporal contexts, while Gen-Omnimatte (Lee et al., 2025) is prone to texture leakage and blurry blobs during subject removal in cluttered backgrounds. These failure cases are filtered out by our quality control pipeline, but they reduce the yield of multi-foreground samples.

Nevertheless, our data engine is fully automated and generalizable—it imposes no assumptions on the underlying decomposition or matting models. As more robust video matting and inpainting methods emerge, our pipeline can be readily applied to construct higher-quality and more diverse layered video datasets.

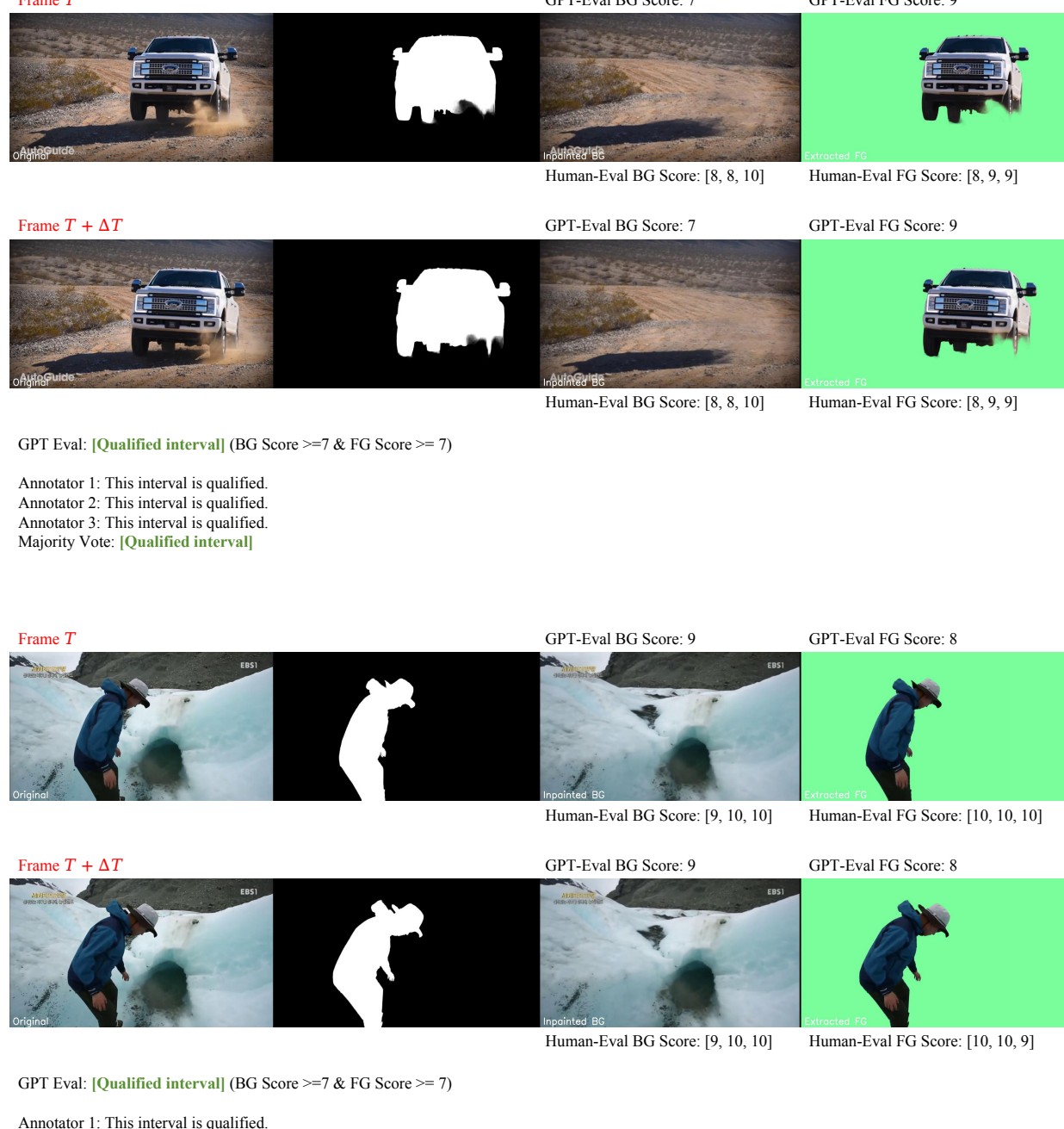

*Figure 13.* **Qualitative analysis of the GPT-4o auto-check.** For each example we show, from left to right, the original frame, the predicted mask, the inpainted background, and the extracted foreground at two endpoint frames ($T$ and $T + \Delta T$), together with the GPT-4o BG/FG scores, the three human scores, and the resulting interval-level decision. GPT-4o agrees with the human majority vote across both kept and rejected intervals, correctly flagging decomposition failures such as residual foreground and background artifacts.

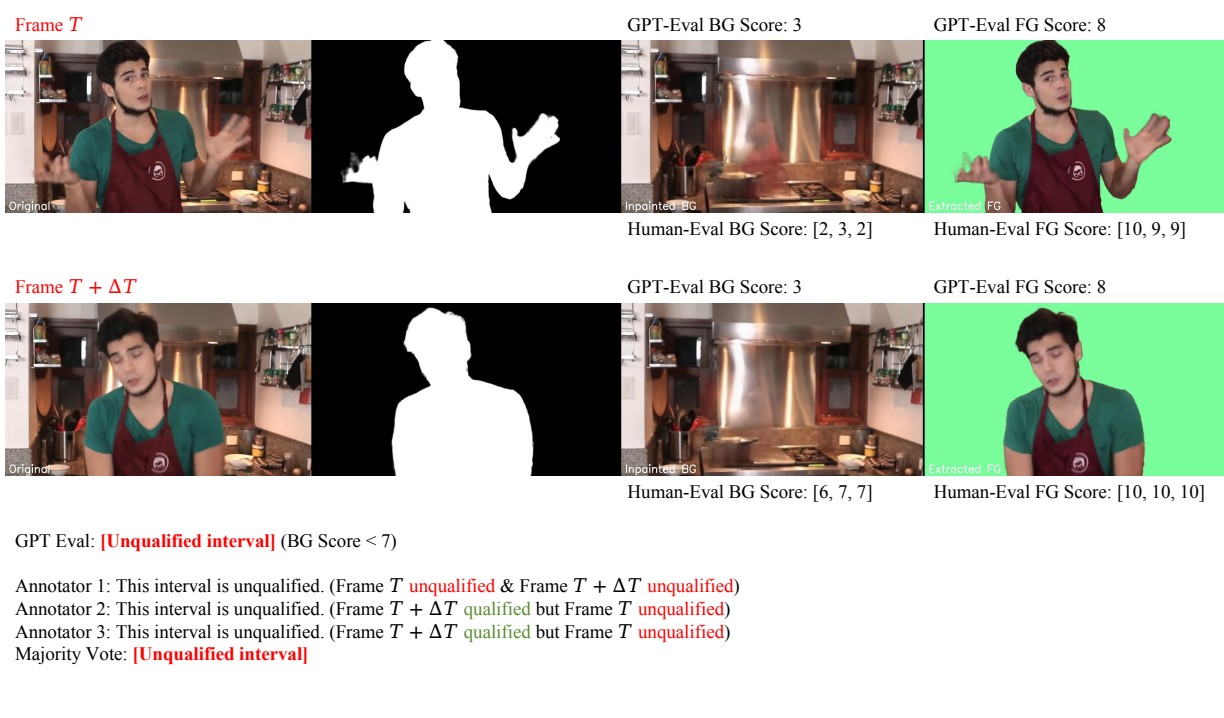

Frame $T$ — GPT-Eval BG Score: 3 — GPT-Eval FG Score: 8

Human-Eval BG Score: [2, 3, 2] — Human-Eval FG Score: [10, 9, 9]

Frame $T + \Delta T$ — GPT-Eval BG Score: 3 — GPT-Eval FG Score: 8

Human-Eval BG Score: [6, 7, 7] — Human-Eval FG Score: [10, 10, 10]

GPT Eval: [Unqualified interval] (BG Score < 7)

Annotator 1: This interval is unqualified. (Frame $T$ unqualified & Frame $T + \Delta T$ unqualified)
Annotator 2: This interval is unqualified. (Frame $T + \Delta T$ qualified but Frame $T$ unqualified)
Annotator 3: This interval is unqualified. (Frame $T + \Delta T$ qualified but Frame $T$ unqualified)
Majority Vote: [Unqualified interval]

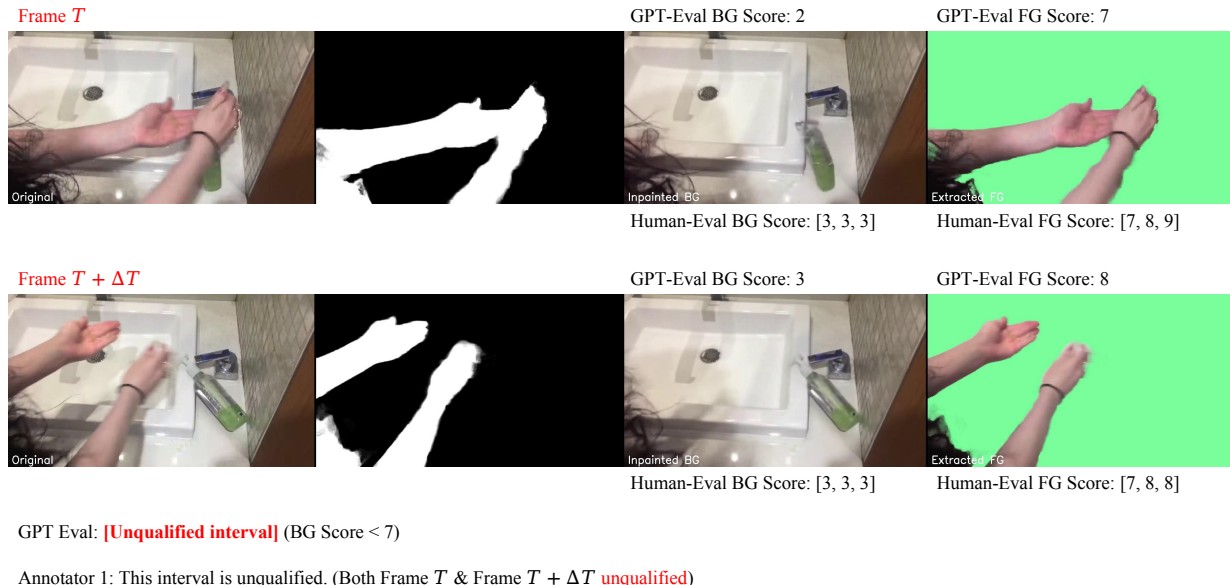

Frame $T$ — GPT-Eval BG Score: 2 — GPT-Eval FG Score: 7

Human-Eval BG Score: [3, 3, 3] — Human-Eval FG Score: [7, 8, 9]

Frame $T + \Delta T$ — GPT-Eval BG Score: 3 — GPT-Eval FG Score: 8

Human-Eval BG Score: [3, 3, 3] — Human-Eval FG Score: [7, 8, 8]

GPT Eval: [Unqualified interval] (BG Score < 7)

Annotator 1: This interval is unqualified. (Both Frame $T$ & Frame $T + \Delta T$ unqualified)
Annotator 2: This interval is unqualified. (Both Frame $T$ & Frame $T + \Delta T$ unqualified)
Annotator 3: This interval is unqualified. (Both Frame $T$ & Frame $T + \Delta T$ unqualified)
Majority Vote: [Unqualified interval]    Hint: the foreground (hair) is not fully removed

*Figure 14.* **Qualitative analysis of the GPT-4o auto-check.** For each example we show, from left to right, the original frame, the predicted mask, the inpainted background, and the extracted foreground at two endpoint frames ($T$ and $T + \Delta T$), together with the GPT-4o BG/FG scores, the three human scores, and the resulting interval-level decision. GPT-4o agrees with the human majority vote across both kept and rejected intervals, correctly flagging decomposition failures such as residual foreground and background artifacts.

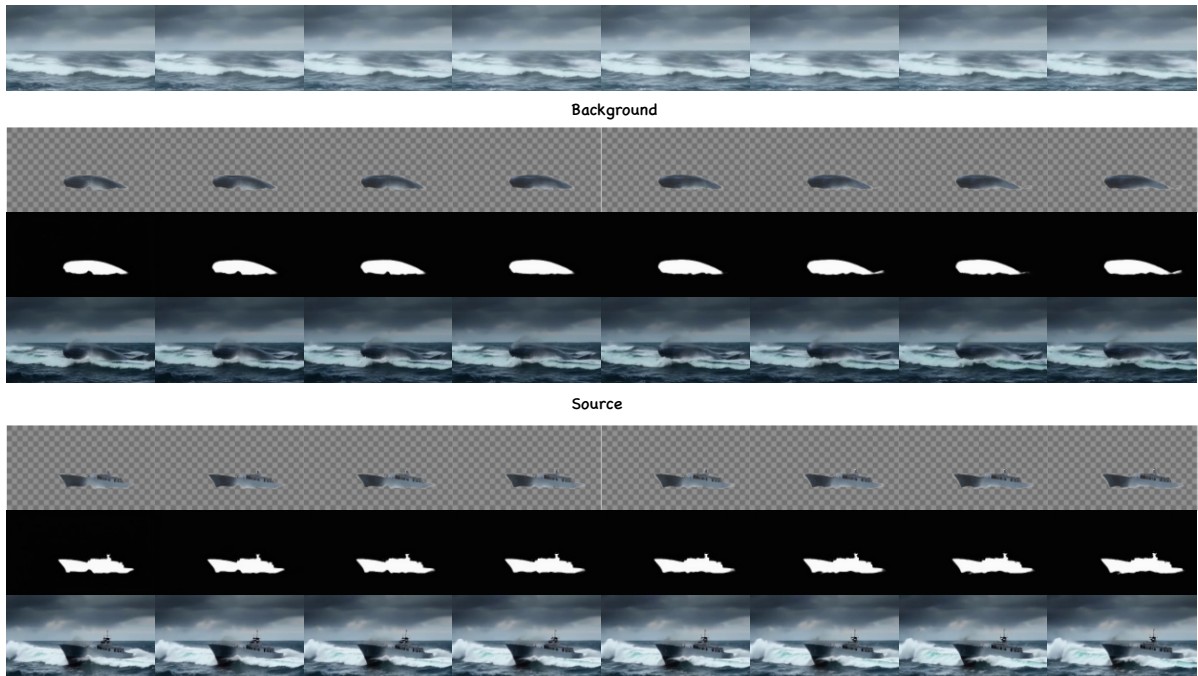

*Figure 15.* **An example of foreground editing using LayerT2V.** We first perform a single sampling and record the state of each layer at each timestep. During editing, we replace the current generated state with the sampled state for irrelevant layers, allowing only the corresponding editing layer to change, thus achieving layered video editing without training.

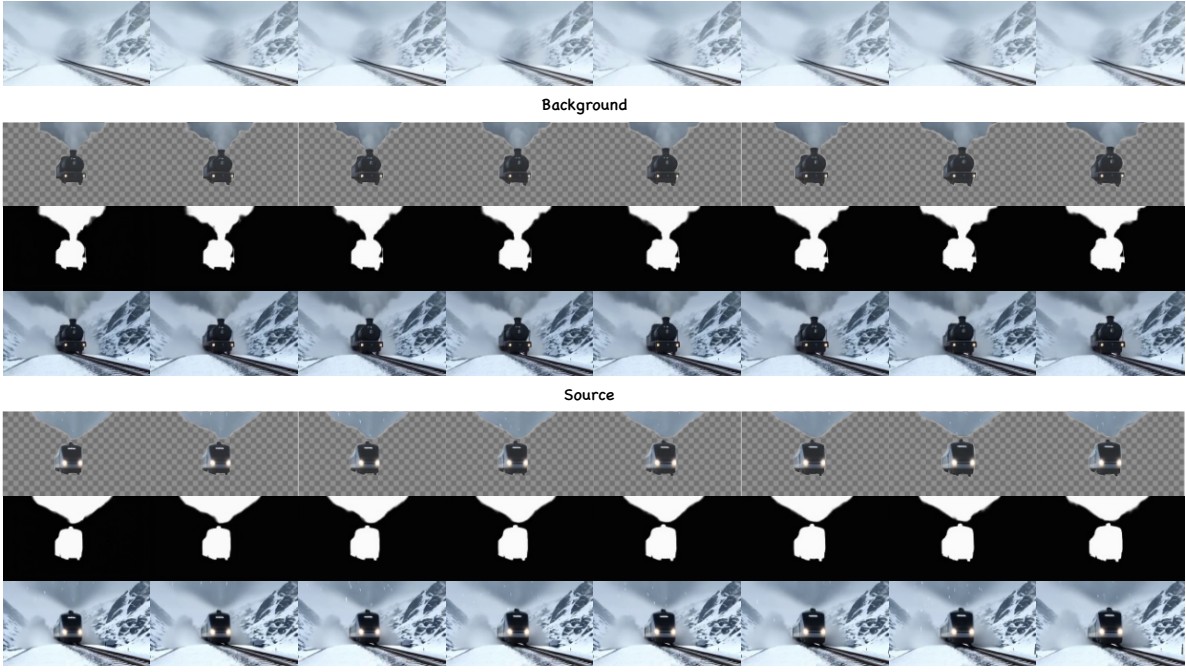

*Figure 16.* **An example of foreground editing using LayerT2V.** We first perform a single sampling and record the state of each layer at each timestep. During editing, we replace the current generated state with the sampled state for irrelevant layers, allowing only the corresponding editing layer to change, thus achieving layered video editing without training.

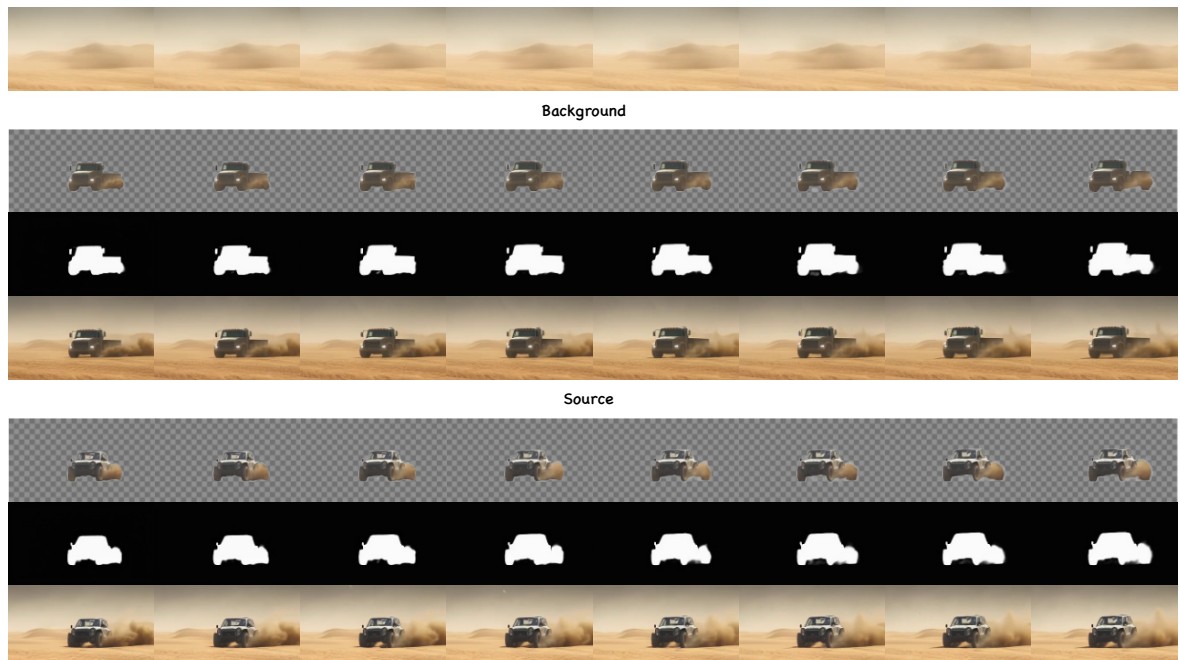

*Figure 17.* **An example of foreground editing using LayerT2V.** We first perform a single sampling and record the state of each layer at each timestep. During editing, we replace the current generated state with the sampled state for irrelevant layers, allowing only the corresponding editing layer to change, thus achieving layered video editing without training.

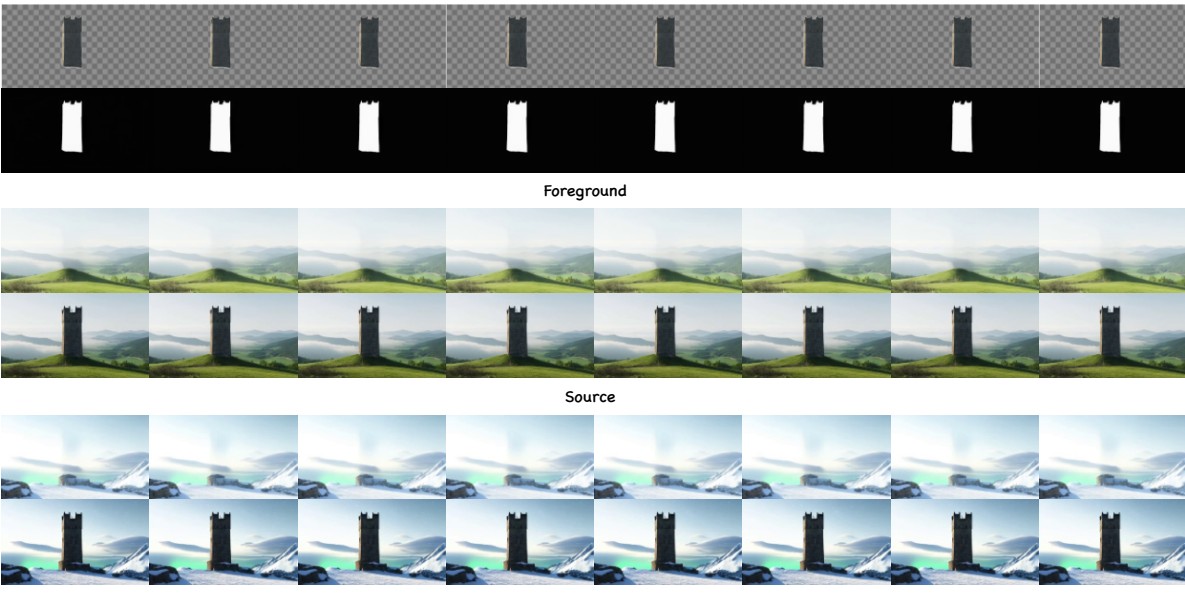

*Figure 18.* **An example of background editing using LayerT2V.** We first perform a single sampling and record the state of each layer at each timestep. During editing, we replace the current generated state with the sampled state for irrelevant layers, allowing only the corresponding editing layer to change, thus achieving layered video editing without training.

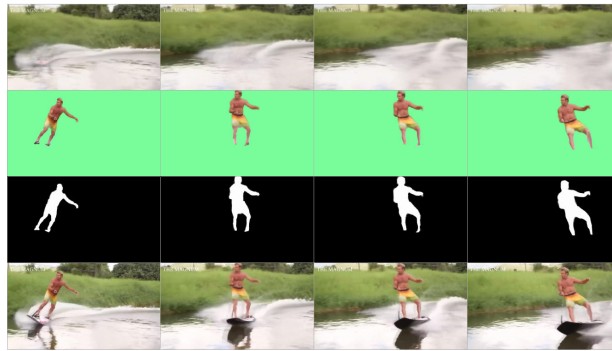

BL: The video shows a man wakeboarding on a river. He is wearing yellow shorts and is being pulled by a boat. The man is seen making a jump and landing back on the water.

FG: A man wakeboarding, wearing yellow shorts, being pulled by a boat, making a jump and landing back on the water.

BG: A river with green grassy banks, calm water with ripples and splashes from the wakeboard.

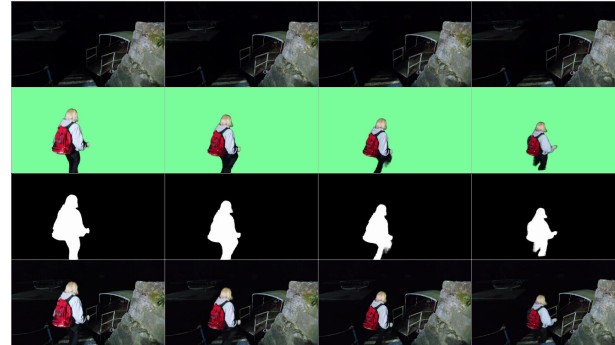

BL: In the video, a woman with blonde hair is seen walking down a set of stairs at night. She is wearing a red backpack and a white jacket. As she reaches the bottom of the stairs ...

FG: A woman with blonde hair wearing a red backpack and a white jacket is stepping onto a small boat with the help of a man in a blue shirt.

BG: A dark night scene at a dock with calm water, a small boat with a white roof, and dim lighting from a few sources in the background.

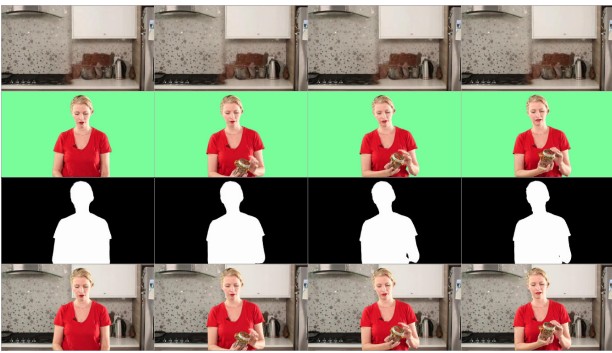

BL: In the video, a woman is standing in a kitchen and talking to the camera. She is wearing a red shirt and has blonde hair. She is holding a jar of pickles in her hands and is ...

FG: A woman with blonde hair wearing a red shirt, holding a jar of pickles and showing it to the camera, standing in a kitchen.

BG: A modern kitchen with white cabinets, a gray backsplash with circular patterns, a stainless steel refrigerator, a stovetop, and a kettle.

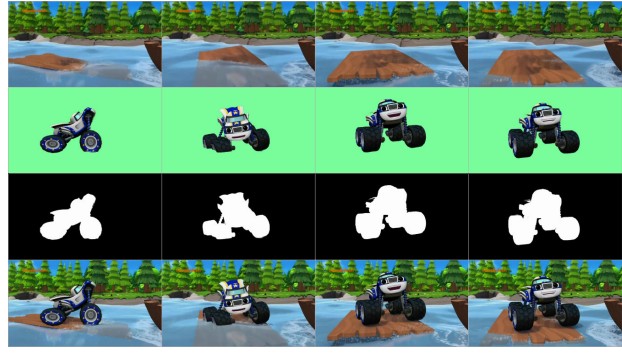

BL: This video features a cartoon monster truck with a blue and white color scheme and stars on its side. The truck is shown driving through a body of water, splashing as it goes ...

FG: A blue and white monster truck with star patterns on its wheels and body, driving on a wooden plank over water, splashing as it moves, with a determined facial expression.

BG: A forest with green trees and a blue sky, a body of water with a wooden plank extending across it.

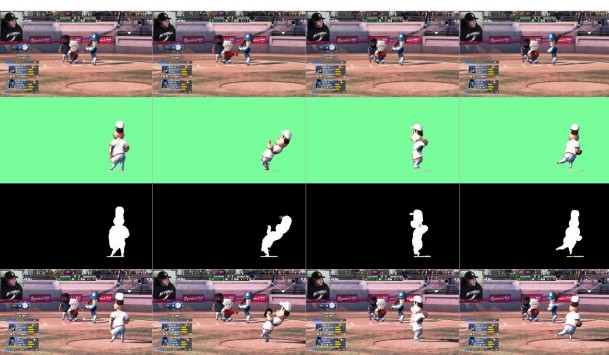

BL: The video is a clip from a baseball video game. It shows a batter at the plate, ready to hit the ball. The pitcher throws the ball, and the batter swings and misses. The catcher ...

FG: A cartoon-style baseball pitcher in a white uniform, standing on the pitcher's mound, preparing to throw the ball.

BG: A baseball stadium with a scoreboard showing the score and game details, spectators in the stands, and a foul ball indicator on the field.

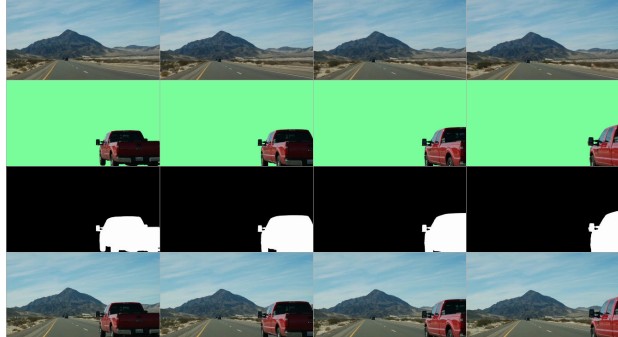

BL: The video is a series of frames taken from a moving vehicle on a highway. The high-way is surrounded by mountains and desert terrain. The sky is clear and blue. The road is ...

FG: A red pickup truck driving on a highway.

BG: A highway surrounded by mountains and desert terrain under a clear blue sky.

*Figure 19.* Here, we present additional visualized samples from our proposed VidLayer dataset. The dataset covers a wide and diverse range of scenarios, including human portraits, sports activities, pets, and outdoor environments. It achieves effective and reliable separation of foreground and background content, thereby providing strong and informative supervisory signals for future research on layered content generation, video decomposition, inpainting, and editing tasks.

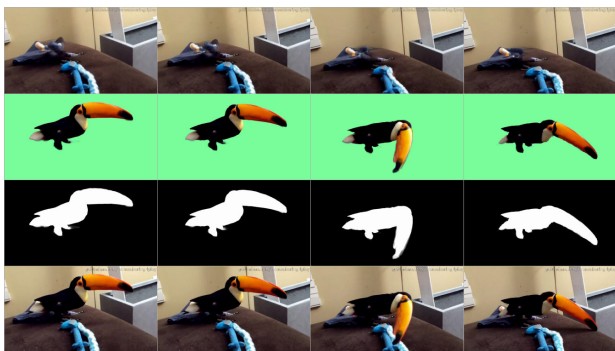

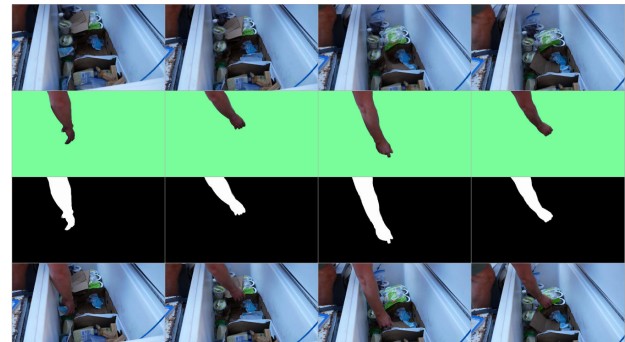

BL: The video shows a toucan standing on a table. The toucan is black and white with an orange beak. There are two stuffed animals on the table, one blue and one white ...

FG: A toucan with a large orange beak and black and white feathers, holding a blue stuffed animal in its beak and moving it around, then picking up a white stuffed animal and ...

BG: A brown table surface with two stuffed animals, one blue and one white, and a beige wall with white blinds in the background.

BL: In the video, a person is seen going through a freezer. The freezer is filled with various food items, including yogurt, cheese, and other packaged goods. The person is wearing ...

FG: A person's arm and hand reaching into a cooler, wearing a pink shirt, picking up and examining various food items including yogurt, cheese, and packaged goods.

BG: A white freezer filled with various food items, located in a kitchen or similar setting.

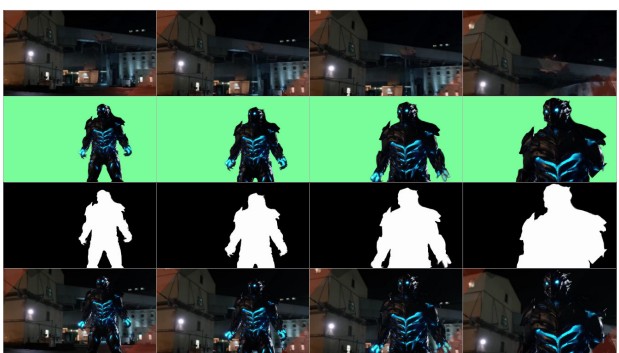

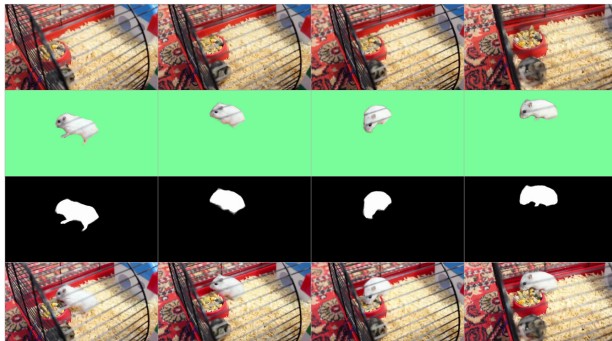

BL: The video shows a humanoid robot standing in a dark alleyway. The robot is made of metal and has blue lights on its chest and hands. It is looking directly at the camera with ...

FG: A humanoid robot with a black metallic suit, glowing blue lines on its chest and hands, and bright blue eyes, standing in a serious pose and looking directly at the camera.

BG: A dark urban alleyway at night, with dimly lit buildings and no other objects or people visible.

BL: The video shows a white hamster and a gray and white hamster in a red cage with a red food bowl filled with seeds. The white hamster is eating from the bowl while the gray and ...

FG: A white hamster with a pink nose eating from a red food bowl filled with seeds, small and fluffy, actively moving its paws to grab the food.

BG: A red cage made of metal bars with a floor covered in wood shavings, a red food bowl filled with seeds, and a patterned red carpet underneath the cage.

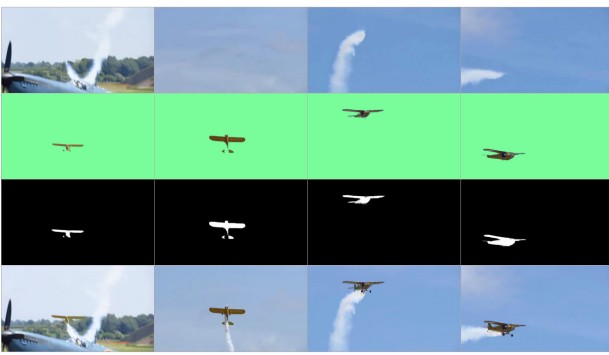

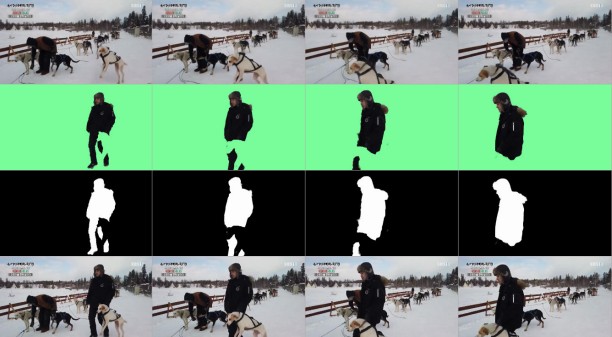

BL: The video shows a yellow airplane with smoke coming out of the back of it. The airplane is flying in the sky, and the smoke is trailing behind it. The airplane is seen from different ...

FG: A yellow airplane performing aerobatics, emitting smoke from the back, seen from different angles in flight.

BG: A clear blue sky with no other objects visible.

BL: The video shows a man in a black jacket and a fur hat standing in the snow with a group of dogs on leashes. The man is seen interacting with the dogs, petting them, and walking ...

FG: A man in a black jacket and a fur hat standing in the snow, interacting with a group of dogs on leashes, petting them, and walking around with them, smiling and laughing, the dogs ...

BG: A snowy landscape with trees and a wooden fence, a vast open snowy field with a few structures in the distance, a clear sky with some clouds.

*Figure 20.* Additional visualized samples from proposed VidLayer dataset.

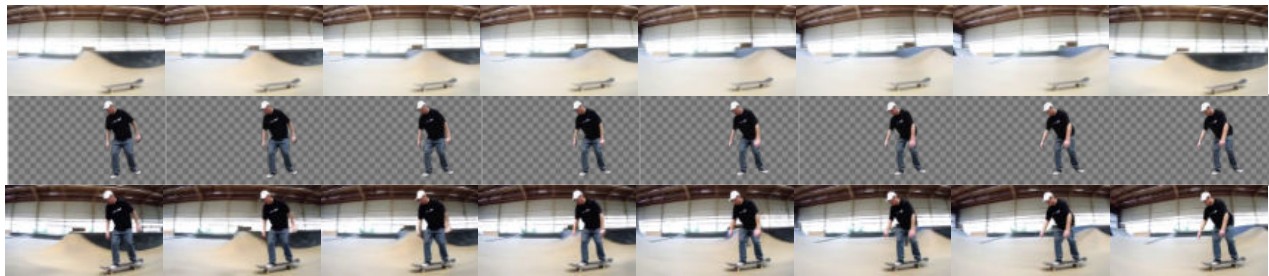

BL: The video shows a man skateboarding in an indoor skate park. He is wearing a black shirt and a white cap. The skateboarder is seen riding down ramps and performing tricks. The park has various ramps and obstacles

FG: A person skateboarding, wearing a black shirt and a white cap, performing tricks and riding down ramps, turning to smile at the camera.

BG: an indoor skate park with various ramps and obstacles, artificial lighting.

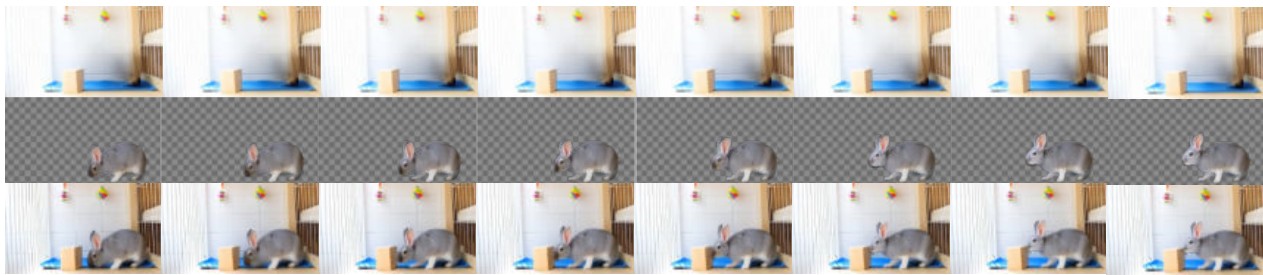

BL: In the video, a grey rabbit is seen inside a cage. The rabbit is standing on a blue mat and there is a wooden block in front of it. The rabbit sniffs the block and then moves away from it. The cage has a white tiled wall and a colorful toy hanging from the top.

FG: a gray rabbit standing on a wooden platform, sniffing a wooden block, with floppy ears and a fluffy coat

BG: a cage with a white tiled wall, a blue mat on the floor, a colorful toy hanging from the top, and a wooden platform

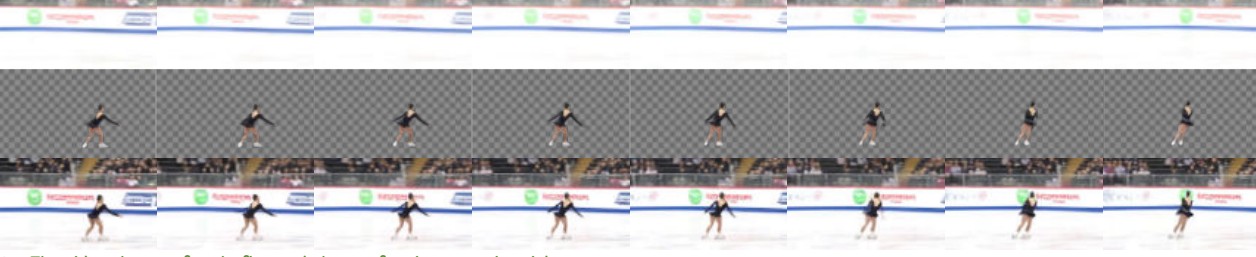

BL: The video shows a female figure skater performing on an ice rink.

FG: A female figure skater wearing a black costume, performing a spin on ice with her arms outstretched, showcasing grace in her movements

BG: An indoor ice rink surrounded by barriers with advertisements and spectators watching from the stands

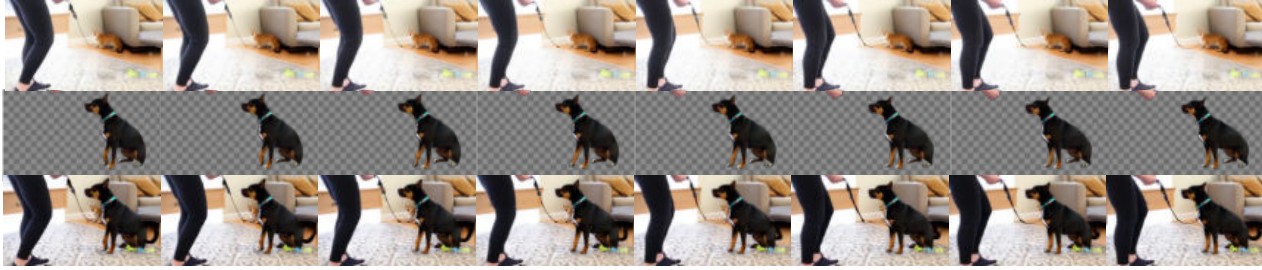

BL: The video shows a dog on a leash being trained by a person.

FG: A black dog wearing a collar and leash, standing and sitting on a patterned rug, being trained by a person.

BG: A living room with a couch, a cat sitting on the floor, and a patterned rug.

*Figure 21.* **Additional qualitative results of LayerT2V.** We show more examples with eight sampled frames per prompt. LayerT2V generates high-fidelity layered outputs, producing clean backgrounds, accurate transparent foregrounds, and coherent recompositions across diverse motions and scenes; the accompanying text provides the full-video description and the layer-specific foreground and background descriptions.

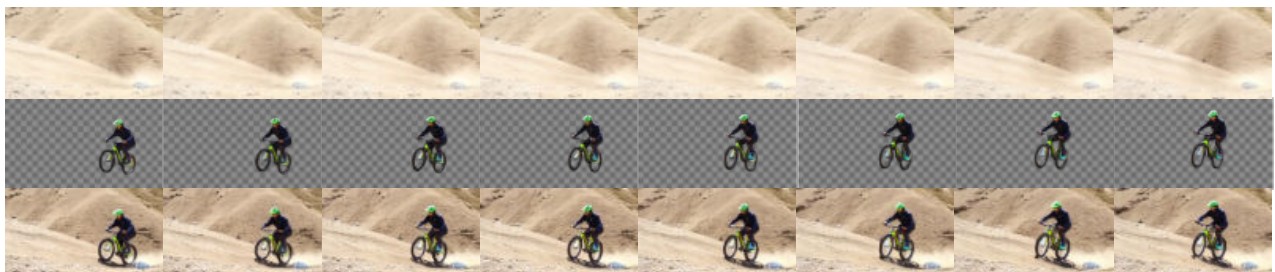

BL: The video shows a man riding a bicycle down a rocky hill. He is wearing a black jacket and a green helmet. The hill is covered in rocks and dirt. The video captures the thrill and excitement of mountain biking, as the riders navigate the challenging terrain with skill and precision.

FG: a cyclist wearing a green helmet and dark clothing, riding a mountain bike at high speed down a rocky hill

BG: a rocky hill covered in loose dirt and scattered stones, with dust being kicked up by the riders

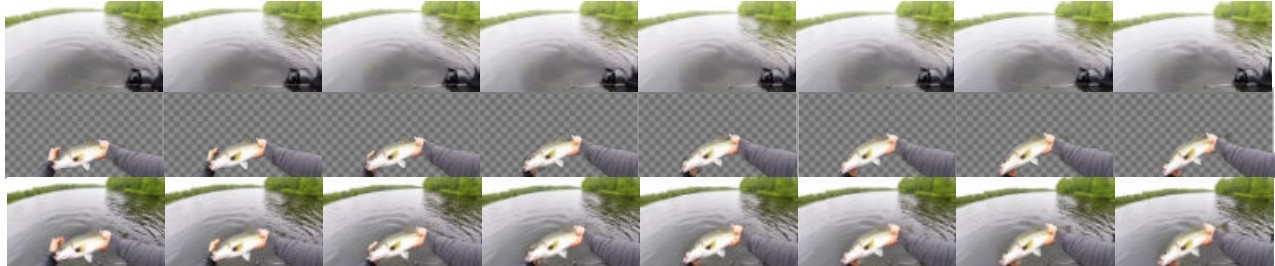

BL: The video shows a man fishing in a lake. He catches a large fish and holds it up for the camera. The man is wearing a gray shirt. The fish is silver with a white belly and appears to be a bass. The man is holding the fish with both hands. The lake is surrounded by trees.

FG: a largemouth bass being held by a person, silver with a white belly, caught on a fishing line, held with both hands, the fish appears to be alive and wriggling slightly

BG: a calm lake surrounded by trees, overcast sky, no other people visible, the water is still with slight ripples

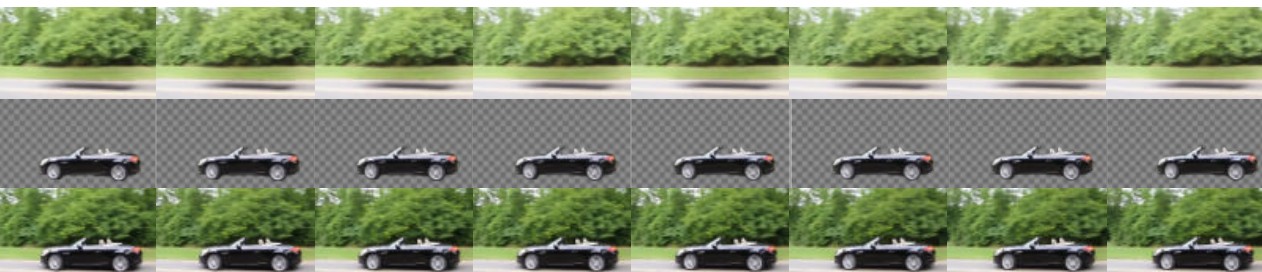

BL: The video shows a man fishing in a lake. He catches a large fish and holds it up for the camera. The man is wearing a gray shirt. The fish is silver with a white belly and appears to be a bass. The man is holding the fish with both hands. The lake is surrounded by trees.

FG: a largemouth bass being held by a person, silver with a white belly, caught on a fishing line, held with both hands, the fish appears to be alive and wriggling slightly

BG: a calm lake surrounded by trees, overcast sky, no other people visible, the water is still with slight ripples

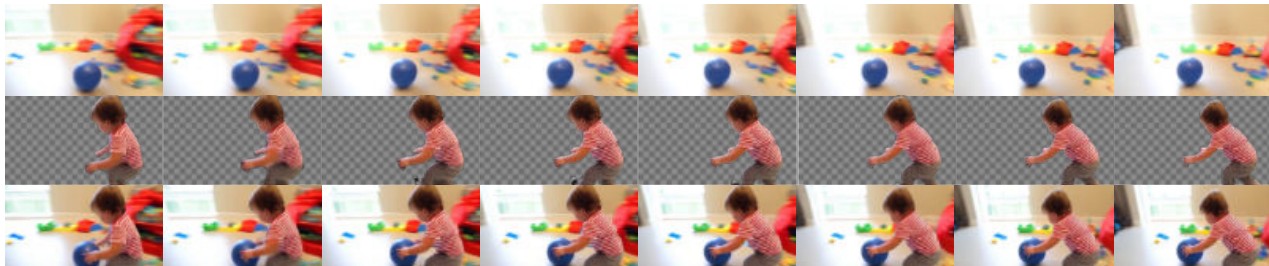

BL: The video shows a man fishing in a lake. He catches a large fish and holds it up for the camera. The man is wearing a gray shirt. The fish is silver with a white belly and appears to be a bass. The man is holding the fish with both hands. The lake is surrounded by trees.

FG: a young child wearing a red and white striped shirt and checkered pants, actively pushing a large blue exercise ball with energetic movements, fully engaged in playful activity

BG: a well-lit indoor play area with various toys scattered around.

*Figure 22.* **Additional qualitative results of LayerT2V.** We show more examples with eight sampled frames per prompt. LayerT2V generates high-fidelity layered outputs, producing clean backgrounds, accurate transparent foregrounds, and coherent recompositions across diverse motions and scenes; the accompanying text provides the full-video description and the layer-specific foreground and background descriptions.

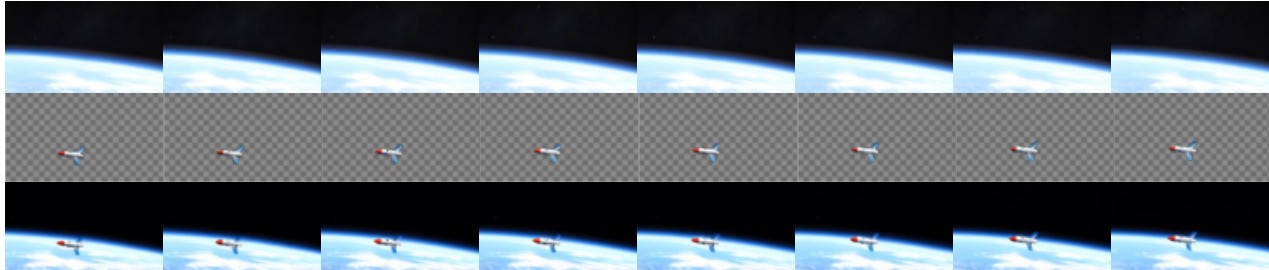

BL: The video features a spacecraft on the space. The spacecraft is then shown lifting off from the ground and ascending into the sky. The spacecraft is white with blue and red markings. The video ends with the spacecraft in orbit around the Earth.

FG: An experimental spaceplane (XS-1) with a sleek white body, blue and red markings. The space shuttle is flying horizontally on Earth.

BG: A clear blue sky with scattered clouds, transitioning into the blackness of space. The curvature of the Earth is visible, with a view of the planet's surface and atmosphere.

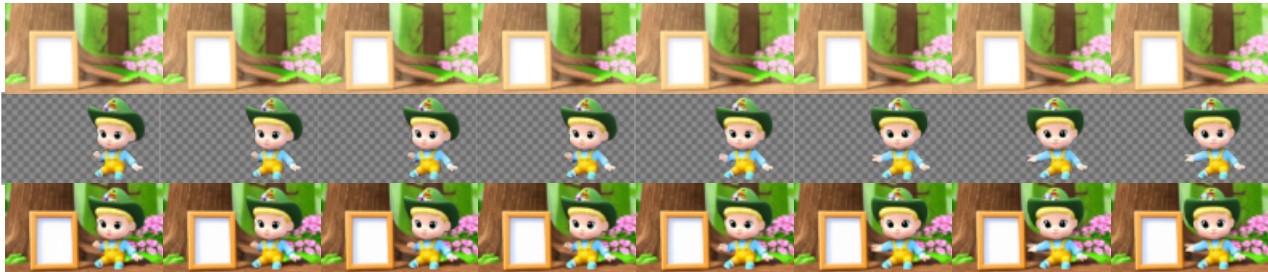

BL: The video shows a cartoon character with blonde hair, wearing a green hat with badges on it, and a blue and yellow outfit. The character is sitting on a log in a forest, with pink flowers visible in the background. The character is seen looking at a picture frame. The character then

FG: a cartoon baby with blonde hair, wearing a green hat adorned with various badges, a blue and yellow shirt, and yellow overalls, sitting on a log, looking at a picture frame, then turning to smile at the camera

BG: a forest setting with tree trunks, green foliage, and pink flowers in the background

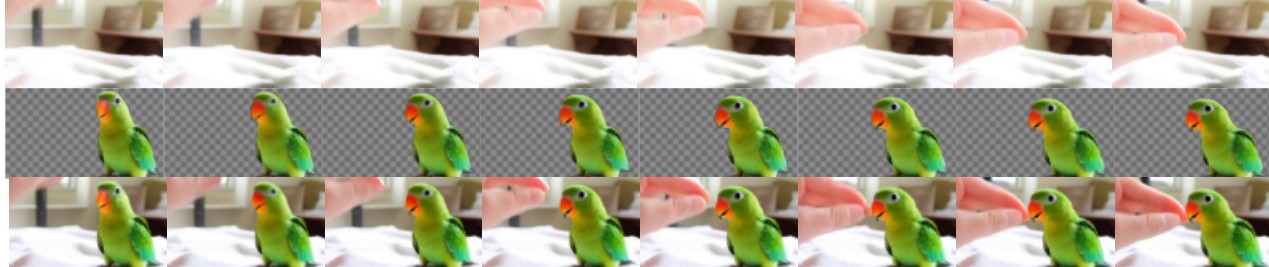

BL: The video shows a close-up of a small green bird with an orange beak. The bird is sitting on a white cloth and being gently petted by a human hand. The bird appears to be a baby parrot. The background is blurred, but it appears to be an indoor setting with a window and

FG: a small green parrot with an orange beak, sitting on a white cloth, being gently petted by a human hand, its feathers fluffed up, appearing calm and content

BG: an indoor setting with a blurred background, including a window and some furniture

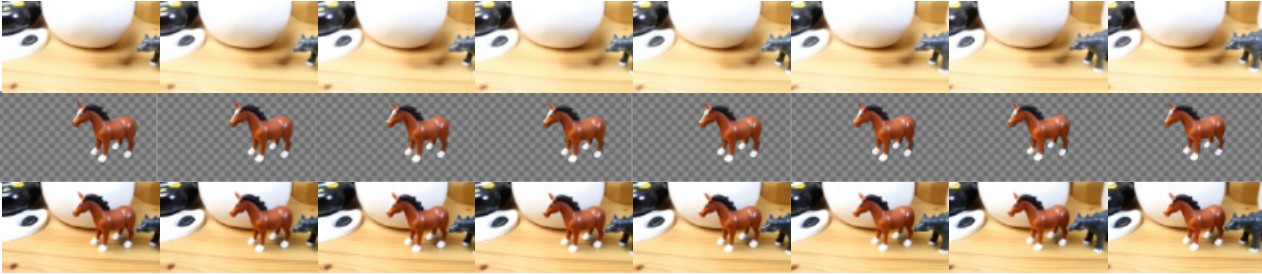

BL: The video shows a series of toy animals, including a horse and a rhinoceros, arranged on a wooden surface. The horse is brown with a black mane, while the rhinoceros is black. The camera pans around the toys, showing them from different angles. The toys appear to be made of

FG: a detailed brown and white toy horse figurine with a black mane, made of plastic, featuring realistic features such as eyes, ears, and hooves, positioned on a wooden surface

BG: a wooden surface with a natural grain pattern, a white object in the background, and other toy animals including a black rhinoceros

*Figure 23.* **Additional qualitative results of LayerT2V.** We show more examples with eight sampled frames per prompt. LayerT2V generates high-fidelity layered outputs, producing clean backgrounds, accurate transparent foregrounds, and coherent recompositions across diverse motions and scenes; the accompanying text provides the full-video description and the layer-specific foreground and background descriptions.

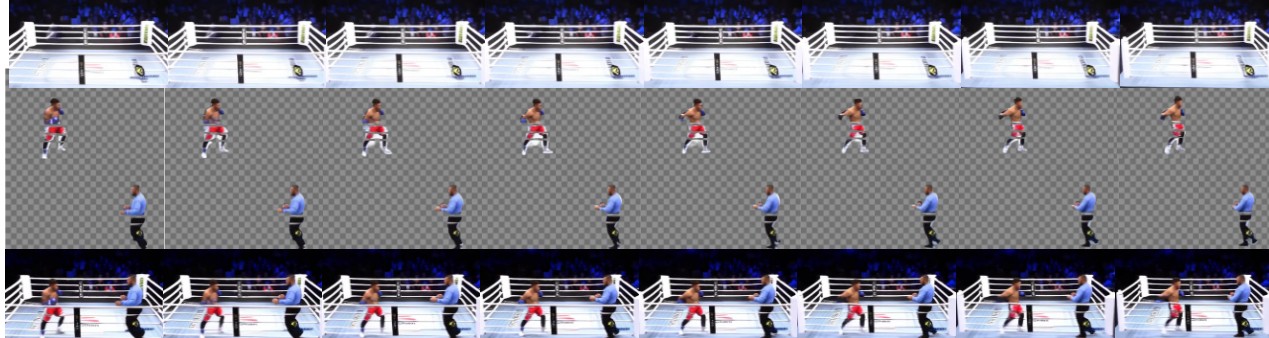

BL: The video shows a boxing match between two men in a ring. One man is wearing red shorts and the other is wearing blue shorts who is trying to defend himself.

FG1: One man is wearing red shorts standing in a boxing

FG2: One man is wearing blue shorts, standing in a boxing

BG: a brightly lit boxing ring with advertisements on the ropes and the mat, surrounded by an audience

*Figure 24.* **Additional two-foreground-layer qualitative results of LayerT2V.** We show more examples with eight sampled frames per prompt. LayerT2V generates high-fidelity layered outputs, producing clean backgrounds, accurate transparent foregrounds, and coherent recompositions across diverse motions and scenes. Extending models to more foreground layers could be a promising future work.

