# OpenReview forum: "LayerT2V: A Unified Multi-Layer Video Generation Framework"
_ICML.cc/2026/Conference — ICML 2026 regular_

### Official Review · Reviewer_fxBq · 2026-03-06

**Soundness:** 3
**Presentation:** 2
**Significance:** 3
**Originality:** 3
**Overall Recommendation:** 4
**Confidence:** 4

**Summary:**

This paper introduces LayerT2V for multi-layer video generation and a large-scale multi-layer video dataset, VidLayer. LayerT2V includes two novel components, Layer Adaptive Normalization (LayerAdaLN) and LayeredCrossAttention. LayerT2V first fine-tunes VAE for alpha mattes. Then, LayerT2V jointly models text tokens, video tokens, and mask tokens. Third, LayerT2V uses LayerAdaLN to inject layer idenity and LayeredCrossAttention to improve layer separation and coherence. For constructing VidLayer, the authors use Qwen3-VL to annotate the descriptions of foreground and background videos, and then use MatAnyone and Gen-Omnimatte to separate foreground and background. Finaly, the authors use GPT-4o to control the quality.

**Compliance With Llm Reviewing Policy:**

Affirmed.

**Final Justification:**

The authors resolved my concerns. Therefore, I raise the rating to weak accept.

**Key Questions For Authors:**

1.  How does LayerT2V define the number of layers?
2. Why do LayerAdaLN and LayeredCrossAttention work? Visualizing attention may be helpful to understand the mechanic.
3. How is the effectiveness of the auto check with GPT-4o?

**Limitations:**

Yes.

**Strengths And Weaknesses:**

Strengths
1. This paper introduces a novel method for multi-layer video generation.
2. The authors construct and claim to release a large dataset, VidLayer. This will benefit the research community.
3. This result is overall sound.

Weaknesses

1. The authors do not conduct experiments to explain why LayerAdaLN and LayeredCrossAttention work, although they do improve the performance in an ablation study.
2. The authors do not analyze VidLayer.
3. The authors do not discuss  the effectiveness of the auto check with GPT-4o.
-  Some parts of the paper are unclear.
1. L194. Missing the definition of LN.
2. The formulation of LayeredCrossAttention (L208-L217) uses $-\inf$ and softmax. Using the Sign function could be better.

---

> ### Author Rebuttal · Authors · 2026-03-31
>
> Thank you for your thoughtful review and the time you dedicated to our work! We sincerely appreciate your recognition of our contributions. Below, we address your specific comments.
>
> > **W1 & Q2**: Why do LayerAdaLN and LayeredCrossAttention work?
>
> **A1**: Thank you for this helpful suggestion. LayerAdaLN addresses the statistical mismatch across serialized layers by assigning each layer its own shift/scale/gate parameters and fusing them with timestep modulation at every block, thereby reducing inter-layer entanglement while preserving the pretrained video prior. The cosine-similarity matrix and L2-norm visualization in **[Figure. 1 (click to view)](https://anonymous.4open.science/r/icml5525-86DD/consin.jpg)** and **[Figure. 2](https://anonymous.4open.science/r/icml5525-86DD/l2.jpg)** show clear layer-specific modulations. This is consistent with the ablation results: LayerAdaLN improves subject consistency and temporal flickering, with a modest gain in text alignment.
>
> LayeredCrossAttention reduces semantic leakage at the cross-attention logit level, thereby improving text alignment and reducing artifacts. In **[Figure. 3](https://anonymous.4open.science/r/icml5525-86DD/layerdcrossattn-illustrate.jpg)** and **[Figure. 4](https://anonymous.4open.science/r/icml5525-86DD/cross-attn-map.jpg)**, vanilla cross-attention shows dense off-target interactions, while Layered Cross-Attention produces sparse routing consistent with the visibility mask, suppressing cross-layer prompt leakage. The added ablation in **[Table. 1 (click to view)](https://anonymous.4open.science/r/icml5525-86DD/cross-attn-table.png)** also shows consistent gains, especially in text alignment.
>
> > **W2**: Analyze of VidLayer.
>
> **A2**: Thank you for the comment. We agree this aspect should have been presented more explicitly. However, the manuscript already analyzes VidLayer beyond construction, including: (i) A scalable data engine with semantic annotation, component extraction, and automatic quality control; (ii) an explicit artifact-checking rubric and temporal filtering rule; (iii) dataset statistics, including 50K clips, ~4M frames, and ~0.6M multi-subject frames; (iv) diversity analysis via scene classification and prompt redundancy; and (v) current limitations and failure modes. We will revise the paper to make these analyses and limitations more explicit.
>
> > **W3 & Q3**: Automatic checking with GPT-4o.
>
> **A3**: Thanks for your question. We will conduct a detailed analysis of its validity: GPT-4o scores 4-panel composites (original, mask, inpainted background, extracted foreground) with an explicit BG/FG rubric that penalizes ghosting, color leakage, floating blobs, and geometric distortions. An interval is kept only if both endpoint key frames satisfy BG >= 7 and FG >= 7.
>
> To validate this rule, we conducted a human study on 100 candidate intervals from the pre-filter pool (50 GPT-kept and 50 GPT-rejected; one interval from each of 100 source videos), using the same $\Delta_{key}$ = 10 as the pipeline. Three annotators, blinded to GPT-4o outputs and to one another, evaluated the same endpoint composites with the same 0–10 BG/FG rubric. Interval labels were assigned by the same endpoint rule and majority vote. For score-level analysis, we compare GPT-4o scores with mean human scores using Spearman correlation.
>
> As shown in **[Table. 2](https://anonymous.4open.science/r/icml5525-86DD/4o-check.png)**, GPT-4o aligns well with human judgment: 100.0% of GPT-kept intervals are also judged keep by humans, 88.0% of GPT-rejected intervals are also judged reject, and overall interval-level agreement is 94.0%. At the score level, GPT-4o correlates well with mean human scores, with Spearman's $\rho$ = 0.96 for BG and 0.90 for FG. We also provide a qualitative analysis of selected evaluation results in **[Figure. 5](https://anonymous.4open.science/r/icml5525-86DD/4o-eval.pdf)**. These examples further support the effectiveness of GPT-4o as a quality filter.
>
> > **W4 Point 1**: Definition of LN.
>
> **A4 (1)**: We will improve the presentation in the revision, including explicitly defining LN as **Layer Normalization** when it first appears in Eq. (6).
>
> > **W4 Point 2**: Softmax and sign function.
>
> **A4 (2)**: Thank you for the suggestion. We use the standard attention mask: invisible positions are set to $-\infty$ before softmax, so they receive zero probability while preserving the relative logits among visible tokens. In contrast, a sign function would distort the original attention computation and is therefore not well suited to this setting.
>
> > **Q1**: Define the number of layers.
>
> **A5**: The number of layers is not predicted automatically. It is predefined by the configured foreground count $K$, with single-foreground generation using [full/blending, bg, fg, $\alpha$] and the general setting supporting up to three foreground-alpha pairs in one denoising pass, i.e., $[full, bg, (fg_i, \alpha_i)]_{i=1}^{K}, K \leq 3$.

---

> > ### Author Rebuttal · Reviewer_fxBq · 2026-04-01
> >
> > The authors resolved my concerns. However, I suggest conducting larger experiments with confidence intervals for the effectiveness of the auto check with GPT-4o.

---

> > > ### Author Response · Authors · 2026-04-04
> > >
> > > Thank you for your positive feedback and valuable suggestions. We sincerely appreciate your recognition that our previous response has sufficiently addressed your concerns. To further validate the effectiveness of the auto-check with GPT-4o, we conduct a larger-scale evaluation with **95% confidence intervals**, examining both its agreement with human judgment and its consistency across repeated runs.
> > >
> > > - **Evaluation Protocol**
> > >
> > >   We expand the human validation set to **300 candidate intervals**, including the original 100 annotated samples and 200 newly added samples, while keeping the sampled videos balanced across dataset categories. We denote the auto-check procedure used in the paper as eval 1, and re-evaluate the same candidate intervals twice under the same protocol, denoted as eval 2 and eval 3. We then compare eval 1 with human annotations, assess consistency across the three repeated evaluations, and further compare models trained on the unfiltered and GPT-4o-filtered datasets under matched data scale and identical settings.
> > >
> > > - **Agreement with Human Judgment**
> > >
> > >     We further **compare eval 1 with human evaluation on the 300 candidate intervals**, including the original 100 annotated samples and 200 newly added ones. Specifically, we compare the keep/reject decisions from eval 1 against human annotations and report the agreement on kept intervals, the agreement on rejected intervals, and the overall agreement; we also report human inter-annotator agreement and the correlation between GPT-4o scores and mean human scores. The results in the table below show **strong agreement between the GPT-4o auto-check and human judgment**, supporting the reliability of the proposed auto-check protocol.
> > >   | Metric | Result | 95% CI |
> > >   |---|---:|---:|
> > >   | eval1-kept intervals also judged keep by humans | 146/150 = 97.3% | [93.3%, 99.0%] |
> > >   | eval1-rejected intervals also judged reject by humans | 133/150 = 88.7% | [82.6%, 92.8%] |
> > >   | Overall agreement | 279/300 = 93.0% | [89.5%, 95.4%] |
> > >   | Human inter-annotator agreement (Fleiss' κ) | 0.910 | --- |
> > >   | BG score Spearman ρ | 0.821 | [0.789, 0.850] |
> > >   | FG score Spearman ρ | 0.858 | [0.832, 0.881] |
> > >
> > > - **Repeated-Evaluation Consistency**
> > >
> > >   Building on this, we compare **eval 1**, **eval 2**, and **eval 3** on the same candidate intervals to assess the consistency of repeated GPT-4o judgments. Specifically, we report the correlation of the assigned BG and FG scores across runs, the overlap of the kept subsets, the agreement on rejected intervals, and the overall label agreement. The table below show strong consistency across repeated evaluations, with high score correlation, high overlap of kept intervals, high agreement on rejected intervals, and near-perfect overall label agreement. These results indicate that GPT-4o is not only **well aligned with human judgment, but also highly consistent across repeated runs**, further supporting its reliability as an automatic quality checker for dataset construction.
> > >   | Pair | BG ρ | BG 95% CI | FG ρ | FG 95% CI | IoU (keep labels) | IoU 95% CI | Keep agreement | Keep 95% CI | Reject agreement | Reject 95% CI | Overall label agreement | Overall 95% CI |
> > >   |---|---:|---|---:|---|---:|---|---:|---|---:|---|---:|---|
> > >   | eval1-eval2 | 0.882 | [0.851, 0.909] | 0.873 | [0.842, 0.900] | 145/151 = 96.0% | [91.5%, 98.2%] | 145/151 = 96.0% | [91.5%, 98.2%] | 146/154 = 94.8% | [90.1%, 97.4%] | 291/300 = 97.0% | [94.4%, 98.4%] |
> > >   | eval1-eval3 | 0.846 | [0.808, 0.879] | 0.861 | [0.828, 0.891] | 147/150 = 98.0% | [94.3%, 99.3%] | 147/150 = 98.0% | [94.3%, 99.3%] | 150/153 = 98.0% | [94.4%, 99.3%] | 297/300 = 99.0% | [97.1%, 99.7%] |
> > >   | eval2-eval3 | 0.831 | [0.792, 0.866] | 0.852 | [0.818, 0.882] | 145/148 = 98.0% | [94.2%, 99.3%] | 145/148 = 98.0% | [94.2%, 99.3%] | 152/155 = 98.1% | [94.6%, 99.3%] | 297/300 = 99.0% | [97.1%, 99.7%] |
> > >
> > > - **Impact on Training Data Quality**
> > >
> > >   To further validate the effectiveness of the auto-check with GPT-4o, we **compare models trained on the original unfiltered dataset and the GPT-4o-filtered dataset under matched data scale and identical settings**. As shown in **[Figure. 6](https://anonymous.4open.science/r/icml5525-86DD/qr-4o.jpg)**, the model trained on unfiltered data exhibits incorrect layer decomposition and visible background artifacts, while the model trained on filtered data produces cleaner and more stable results. The quantitative comparison in the table below shows consistent gains across Subject Consistency, Temporal Flickering, and Text Alignment, further confirming the effectiveness of the GPT-4o auto-check (Triplets are reported in the order foreground / background / blended).
> > >   | Configuration | Subject Consistency | Temporal Flickering | Text Alignment |
> > >   |---|---|---|---|
> > >   | w/o auto-check | 0.959 / 0.968 / 0.972 | 0.955 / 0.967 / 0.961 | 0.172 / 0.219 / 0.211 |
> > >   | **w/ auto-check** | **0.983 / 0.984 / 0.975** | **0.987 / 0.983 / 0.968** | **0.201 / 0.231 / 0.214** |

---

### Official Review · Reviewer_BuM5 · 2026-03-07

**Soundness:** 3
**Presentation:** 3
**Significance:** 3
**Originality:** 3
**Overall Recommendation:** 4
**Confidence:** 3

**Summary:**

This paper proposes LayerT2V, a unified multi-layer autoregressive framework for text-to-video generation that models video tokens at multiple semantic levels to improve temporal coherence and generation quality. The approach introduces a hierarchical token prediction strategy and demonstrates competitive performance on several video generation benchmarks compared to existing autoregressive and diffusion-based methods.

**Compliance With Llm Reviewing Policy:**

Affirmed.

**Final Justification:**

This paper presents a layer-based text-to-video generation framework, which decomposes the video synthesis process into separate layers to enable more controllable and high-quality generation. The approach is both novel and well-motivated. Overall, I am inclined to accept this work.

**Key Questions For Authors:**

1. It would be better to further illustrate the quality control/verification process.
2. It would be more convincing to show cases for longer video generation, multi-subjects video generation, and video editing.

**Limitations:**

Yes

**Strengths And Weaknesses:**

# Strength:
1. The paper is clearly written and well structured, with a coherent presentation of the proposed framework and its motivation.
2. The experimental section is reasonably comprehensive and demonstrates consistent improvements over several baselines, providing useful insights into hierarchical autoregressive modeling for video generation.

# Weakness:
1. The quality control process in the dataset construction could be described more clearly. The paper mentions using MatAnyone and Gen-Omnimatte for automatic matting and generation, but it is unclear how the final quality of generated videos is verified. In particular, if GPT-4o or other vision-language models are used for filtering, the protocol should be clarified, such as whether frames are sparsely sampled for evaluation or whether full video sequences are checked.
2. The scalability of the proposed architecture is also somewhat underexplored. While the method performs well on the presented benchmarks, it remains unclear how the framework behaves under more challenging scenarios such as longer video generation or scenes containing many interacting subjects.
3. Finally, although the paper focuses on generation quality, the evaluation of editing capabilities is limited. More editing cases and analysis would help better demonstrate the controllability and robustness of the system in practical scenarios.

---

> ### Author Rebuttal · Authors · 2026-03-31
>
> Thank you for your thoughtful and positive feedback on our work! We greatly appreciate the reviewer's careful review and approval of our framework. Below, we provide clarifications and additional experiments that address the concerns you raised.
>
> > **W1 & Q1**: Quality control process.
>
> **A1**: Thank you for the suggestion. We agree that the quality-control protocol should be made more explicit in the main paper. We do in fact specify this process in Appendix C.2, but we acknowledge that it may not have been sufficiently visible.
>
> Concretely, the final verification is performed by a structured key-frame-based checker rather than dense full-sequence inspection. For each sampled key frame, we build a 4-panel composite consisting of the original frame, object mask, inpainted background, and extracted foreground. GPT-4o then outputs two explicit scores, BG Score and FG Score, under a fixed rubric that checks artifacts such as ghosting/color leakage, floating blobs, and foreground degradation.
>
> Temporal validation is done at the interval level: with $\Delta_{key}=10$, an interval $[N, N+\Delta_{key}]$ is kept only if both endpoint frames satisfy BG $\ge 7$ and FG $\ge 7$; otherwise it is rejected. We also illustrate accepted/rejected examples in Fig. 8. For the effectiveness analysis of the validation process, please refer to **A3** of Reviewer fxBq's response for details, where the results are presented in **[Figure. 1 (Click to view)](https://anonymous.4open.science/r/icml5525-86DD/4o-eval.pdf)**.
> We will revise the main text to state this protocol more clearly.
>
>
> > **W2 & Q2**: Scalability to more complex scenarios and long video generation.
>
> **A2**: We thank the reviewer for raising this important point. We agree that the scalability of LayerT2V to more complex scenes and longer temporal horizons should be clarified more explicitly.
>
> For more complex scenes, the current submission already provides initial empirical evidence beyond simple single-subject scenarios. LayerT2V is not limited to the simplest setting:
>
> - VidLayer includes a dedicated multi-subject subset.
> - In Stage 3, we extend the same framework to the multi-foreground setting by serializing additional foreground-mask pairs.
> - Figure 4 in the main paper already illustrates three generation modes, including single-foreground with multiple subjects and multi-foreground joint generation.
> - Appendix Figure 15 further shows a joint generation result with multiple foreground layers and more complex subject interactions.
>
> Taken together, these results indicate that our method has already been trained on and demonstrated in multi-subject scenarios, rather than remaining only a conceptual extension. In the revised version, we will add more qualitative examples involving richer interactions and will more clearly distinguish different multi-subject settings.
>
> For longer videos. We agree that the current submission does not yet include a dedicated long-horizon benchmark, as our reported training and inference settings are primarily based on 41-frame videos for controlled evaluation. That said, the LayerT2V formulation itself is not intrinsically tied to short videos. Specifically, we serialize the latents of different layers along the temporal dimension while preserving the input structure expected by the pretrained video backbone, which allows us to reuse its temporal modeling capacity. In addition, LayerAdaLN and layer-aware cross-attention are conditioning modules defined over layer identity rather than a fixed frame count.
>
> > **W3 & Q2**: Explain of editing capabilities
>
> **A3**: Thank you for the valuable suggestion. We agree that video editing is an important practical application of layered video generation. Our key point is that LayerT2V produces, in a single pass, the structured representations needed for editing: a temporally aligned full video, a background layer, foreground RGB layers, and their corresponding alpha masks. This layered formulation directly supports two practical editing modes: background-only editing, where the background is replaced or modified while preserving the foreground appearance and motion; and foreground-only editing, where the background remains unchanged and only the target foreground is edited.
>
> We show several LayerT2V-based video editing examples in **[Figure. 2](https://anonymous.4open.science/r/icml5525-86DD/edit-case.pdf)**, which better illustrate the editability and controllability enabled by its layered representations. In the revision, we will further analyze and discuss LayerT2V’s editing capability and include more editing results.

---

> > ### Author Rebuttal · Reviewer_BuM5 · 2026-04-03
> >
> > I really appreciate the authors' efforts during the rebuttal, which has fully addressed my concerns, I will increase my rating acoordingly.
> >
> > In addition, while the current demos is a little bit simple, I think adding more results on video editing would help to improve the impact of this work.

---

> > > ### Author Response · Authors · 2026-04-04
> > >
> > > Thank you for your positive feedback and for your thoughtful suggestions. We sincerely appreciate your recognition that our response has fully addressed your concerns.
> > >
> > > We also greatly appreciate your helpful suggestion regarding the video editing results. As shown in **[Figure. 3](https://anonymous.4open.science/r/icml5525-86DD/more-edit-case.pdf),** we have further added several editing examples to better demonstrate the editing capability of LayerT2V. These additional results show that the learned layered representation can support flexible video editing while preserving temporal coherence and controllability.
> > >
> > > In the revised version, we will present the editing ability of our method more comprehensively and provide a more detailed analysis of this capability.
> > >
> > > Your feedback has been very helpful in improving and strengthening our paper. Thank you again for your valuable feedback and careful review.

---

### Official Review · Reviewer_jdkN · 2026-03-09

**Soundness:** 2
**Presentation:** 3
**Significance:** 3
**Originality:** 3
**Overall Recommendation:** 5
**Confidence:** 3

**Summary:**

The paper introduces LayerT2V, a multi-layer video generation framework that produces a full video, background, foreground, and alpha masks in a single pass. Its key idea lies in serializing multiple layers along the temporal dimension, and it addresses layer identity ambiguity and semantic leakage through two modules: LayerAdaLN and layered cross-attention. The authors also introduce VidLayer, a large-scale multi-layer video dataset. Experiments demonstrate that the method significantly outperforms existing techniques in visual quality and cross-layer consistency.

**Compliance With Llm Reviewing Policy:**

Affirmed.

**Final Justification:**

The experimental analysis in this work is thorough, and the method makes sense. I believe it can advance the community's progress on layer-video generation. Given that the authors addressed most of my concerns in the rebuttal, I lean toward accepting the paper.

**Key Questions For Authors:**

1.In Eq.4, after concatenating the full video, background, foreground, and mask along the temporal dimension, how should RoPE be configured (or is RoPE even needed)?  The authors conducted an ablation study on 4D-RoPE in the paper. Could you elaborate on the detail, because to prove that 4D-RoPE is problematic, its implementation must be aligned with LayerAdaLN.

2. Table 3 shows that the combination of both modules yields the best results. However, has there been any analysis of what each module is primarily responsible for? Does LayerAdaLN contribute more to temporal stability, while Layer Cross-Attention contributes more to text alignment? Could a more fine-grained decomposition be provided? Are there experimental results for the baseline + Layer Cross-Attention?

3. Could some failure cases be added?

**Limitations:**

Yes.

**Strengths And Weaknesses:**

**Strength**:

1.The presentation in the paper are clear. For example, Fig.1 clearly illustrates the model architecture and training pipeline, while Fig.2 clearly presents the dataset construction pipeline, making it easy for readers to understand.

2.Although layered cross-attention and AdaLN are not novel in themselves, applying them together to the task of layered video generation and achieving strong results is commendable. The experience of how to combine these two modules (such as sharing the same initial noise across different layers) is valuable.

3.An efficient data collection pipeline is proposed, and the authors commit to open-sourcing it, which is significant for the development of the entire community.

---

**Weakness**:

1.My primary concern is with the ablation study, particularly the insufficient experimental details regarding 4D-RoPE. For example, scaling to the generation of multiple layers seems limited compared to 4D-RoPE. Furthermore, relying solely on LayerAdaLN appears unable to express depth relationships, which could become a bottleneck when multiple foreground layers are present.

2.For Eq. 10, is this an overly idealistic assumption that all scenes follow standard alpha compositing? However, in scenarios such as semi-transparent objects or light-emitting objects, the model being coerced into fitting this simplified assumption may lead to unrealistic results.

---

> ### Author Rebuttal · Authors · 2026-03-31
>
> We sincerely thank the reviewer for the thorough and highly constructive evaluation! We greatly appreciate the recognition of our contributions and address all raised concerns below.
>
> > **W1 & Q1**: RoPE configuration and 4D-RoPE ablation.
>
> **A1**: Thank you for these valuable questions. RoPE is still used after temporal concatenation: we apply native 3D RoPE independently within each serialized layer segment to preserve intra-layer spatiotemporal relative positions, and add a fixed (non-learned) per-layer offset to disambiguate layer identity, preventing positional confusion after concatenation.
>
> - **4D-RoPE ablation:** For a fair comparison, we only replace the native 3D positional embedding (42,42,44) with a 4D (38,38,40,12) variant while keeping LayerAdaLN and LayeredCrossAttention. The performance drop can be attributed to two factors: (1) reallocating dimensions from spatiotemporal axes to the layer axis weakens spatiotemporal modeling and misaligns with the pretrained phase relationships; (2) our layer indices correspond to heterogeneous semantic slots rather than continuous geometric depth, so modeling them as a continuous positional axis introduces inappropriate proximity priors. To provide a more detailed ablation analysis, we quantitatively evaluate two 4D-RoPE configurations in **[Table. 1 (click to view)](https://anonymous.4open.science/r/icml5525-86DD/rope-table.png)**: both variants notably degrade subject consistency and temporal flickering, further confirming the above analysis.
>
> - **Depth and Scalability:** LayerAdaLN is not designed to model layer indices as continuous physical depth, but to inject layer identity and layer-specific statistics into the shared DiT. In our task, different layers are semantically heterogeneous components with distinct statistics, rather than regularly sampled depth slices, so treating layer as a continuous positional dimension is not the most appropriate inductive bias. Layer information is jointly modeled by two complementary mechanisms: (i) 3D spatiotemporal priors preserved through temporal serialization with fixed per-layer RoPE offsets, and (ii) explicit layer-specific modulation via LayerAdaLN. Through these two mechanisms, we can naturally extend to more foreground-mask pairs.
>
> > **W2**: Alpha Compositing.
>
> **A2**: Thank you for raising this important point. Eq. (10) follows standard alpha compositing and remains valid for semi-transparent objects and soft alpha, but not strictly for effects such as self-emission. Therefore, our method can still generate semi-transparent objects, as illustrated by the transparent glass example in **[Figure. 1 (click to view)](https://anonymous.4open.science/r/icml5525-86DD/trans.png)**. More importantly, Eq. (10) serves only as a low-weight auxiliary consistency term in latent space, while the primary training objective remains the flow-matching loss in Eq. (9), so it should be understood as a soft cross-layer consistency prior rather than a hard compositing constraint. We will clarify this scope in the revision.
>
> > **Q2**: Main Contribution of each module.
>
> **A3**: Thank you for this suggestion. Our analysis confirms the reviewer's intuition: LayerAdaLN and LayeredCrossAttention play complementary but distinct roles:
> - LayerAdaLN mainly addresses statistical heterogeneity across layers via per-layer modulation, thereby improving layer separation, subject consistency, and temporal stability. The ablation shows consistent gains in subject consistency and temporal flickering, with a smaller gain in text alignment.
> - LayeredCrossAttention mainly improves text alignment by suppressing cross-layer semantic leakage through layer-wise visibility constraints. As shown in **[Figure. 2](https://anonymous.4open.science/r/icml5525-86DD/cross-attn-map.jpg)**, vanilla cross-attention exhibits chaotic interactions, whereas LayeredCrossAttention restricts attention to the designated prompts. The additional ablation below further supports this (Triplets are reported in the order foreground / background / blended):
>
> | Configuration | Subject Consistency | Temporal Flickering | Text Alignment |
> |---|---|---|---|
> | Baseline | 0.931 / 0.942 / 0.924 | 0.955 / 0.961 / 0.945 | 0.169 / 0.182 / 0.188 |
> | Baseline + LayeredCrossAttention | **0.941 / 0.953 / 0.938** | **0.964 / 0.969 / 0.953** | **0.191 / 0.218 / 0.206** |
>
> > **Q3**: Failure cases.
>
> **A4**: Thank you for the suggestion. As shown in **[Figure. 3](https://anonymous.4open.science/r/icml5525-86DD/failure-case.png)**, we will add a failure case where cross-layer consistency is preserved but the model produces saturated alpha channels and absorbs transmission/refraction effects into the premultiplied foreground RGB, reducing decomposition independence and downstream editing fidelity.

---

> > ### Author Rebuttal · Reviewer_jdkN · 2026-04-01
> >
> > Thank you for your response. My concerns have been properly addressed. I will therefore raise my score accordingly.

---

> > > ### Author Response · Authors · 2026-04-04
> > >
> > > Thank you for your feedback and for the constructive review process. We sincerely appreciate your time, consideration, and valuable insights. We are glad that our clarifications addressed your concerns, and we truly appreciate your recognition of our work.

---

### Official Review · Reviewer_pNzn · 2026-03-11

**Soundness:** 2
**Presentation:** 3
**Significance:** 2
**Originality:** 3
**Overall Recommendation:** 3
**Confidence:** 4

**Summary:**

The paper introduces LayerT2V, presenting a unified layered representation approach for text-to-video generation to better decouple complex scene dynamics, separating foreground subjects from background environments.

**Compliance With Llm Reviewing Policy:**

Affirmed.

**Final Justification:**

I thank the authors for the detailed rebuttal, which clarifies the implementation of temporal serialization and the associated computational trade-offs.

While I acknowledge the solid technical execution and the value of the VidLayer dataset, the core methodology remains, in my view, a targeted architectural extension of existing DiT paradigms rather than a fundamental algorithmic breakthrough. While the work has clear empirical utility, its conceptual novelty feels incremental for a venue like ICML. I will therefore maintain my score to reflect this balance between its practical merits and its limited conceptual leap.

**Key Questions For Authors:**

1. How does the architecture prevent ghosting artifacts when foreground objects undergo rapid, non-rigid deformations that heavily occlude the background layer?

2. What is the exact computational overhead (FLOPs and memory footprint) of processing multiple layers compared to a standard flattened video-tubelet approach?

**Limitations:**

The paper does not adequately address the failure modes of layered representations, particularly the severe artifacting and identity breakdown during complex temporal occlusions.

**Strengths And Weaknesses:**

Strengths:

1. The conceptual decoupling of video generation into layered representations is a logical approach to improving spatial controllability.

Weaknesses:

1. Layered representations in latent diffusion spaces notoriously struggle with complex occlusions, often leading to structural tearing or ghosting artifacts. The paper lacks a rigorous solution to maintain temporal consistency across these isolated layers over long horizons.

2. Propagating multiple spatial layers simultaneously through a multimodal diffusion transformer introduces severe computational scaling issues. The empirical evaluation does not sufficiently justify this extreme computational overhead compared to the marginal gains in video fidelity.

---

> ### Author Rebuttal · Authors · 2026-03-31
>
> We sincerely thank the reviewer for the thorough and highly constructive evaluation! We have carefully considered all questions raised and provide our clarifications and responses below.
> > **W1 & Q1**: Ghosting / temporal consistency under heavy occlusion.
>
> **A1**: Thank you for this important concern. LayerT2V does not propagate layers independently and align them afterward. Instead, it serializes background, foreground, and alpha latents along the temporal dimension and jointly denoises them on a shared trajectory with shared noise, making cross-layer consistency an intrinsic property of generation.
>
> To reduce ghosting and structural mismatch under challenging motions, our method includes three explicit mechanisms:
> 1. LayerAdaLN addresses the large statistical gap across layers, reducing feature entanglement from naive shared normalization;
> 2. Layered Cross-Attention suppresses cross-layer semantic leakage via layer-wise text visibility;
> 3. Compositing consistency loss and mask reconstruction loss directly regularize cross-layer coherence and mask boundary stability.
>
> LayerT2V produces clean foregrounds, temporally stable mattes, and complete backgrounds, while improving temporal flickering and subject consistency over LayerFlow. Table 3 verifies the contribution of both LayerAdaLN and Layered Cross-Attention.
>
> To more directly address the reviewer’s concern, we additionally provide targeted examples in **[Figure. 1 (click to view)](https://anonymous.4open.science/r/icml5525-86DD/deformation.png)**: in (a), the foreground occludes a large portion of a cluttered background, yet the generated background remains temporally stable without noticeable artifacts; in (b), under rapid non-rigid deformation and severe foreground-background interaction, the background remains temporally coherent with no obvious ghosting or structural tearing, even across long temporal horizons. These results directly show that LayerT2V maintains cross-layer consistency while suppressing artifacts.
>
> > **W2**: Discussion on the issue of expenses.
>
> **A2**: We agree that jointly modeling multiple video layers introduces additional cost. Compared with complex alignment modules, however, LayerT2V adopts a simple and effective design that achieves strong cross-layer alignment with acceptable overhead, while remaining model-agnostic and extensible to downstream tasks such as video editing:
> 1. **Cross-layer alignment.**
> The main advantage of temporal serialization is not only visual quality, but also preserving cross-layer consistency, a core challenge in multi-layer generation. More complex alignment pipelines may save computation, but do not necessarily yield better consistency. By temporally concatenating layers, LayerT2V makes consistency intrinsic to generation, allowing the model to leverage the backbone’s temporal modeling ability during training.
> 2. **Model-agnostic design.**
> Complex alignment modules are often model-specific and harder to generalize across backbones. In contrast, temporal serialization avoids such components, making LayerT2V compatible with a broad class of DiT-based video generation models.
> 3. **Extensibility.**
> Because structured layered representations are built directly into generation, LayerT2V can be adapted more easily to applications such as video editing without major architectural changes. **[Figure. 2](https://anonymous.4open.science/r/icml5525-86DD/edit-case.pdf)** shows editing examples demonstrating that the learned layered representations support practical editing while preserving temporal coherence and layer controllability.
>
> As video tokenizers and VAEs achieve stronger spatiotemporal compression, latent-space layer serialization becomes increasingly practical. We believe such architectural exploration should be encouraged rather than dismissed.
>
> > **Q2**:  What is the exact computational overhead?
>
> **A3**: Thank you for this important question. We report the latest empirical inference cost below. Since LayerT2V jointly generates four aligned outputs in a single pass, reporting only wall-clock runtime does not fully reflect its effective generation throughput. We therefore additionally report Effective Output FPS (counting all generated frames across aligned outputs). On a single H100 GPU, LayerT2V generates four aligned layers with 161 total frames at 672×384 using 24.4 GB memory and and a runtime of 157s. For reference, the original Wan 2.1 1.3B does not support 672×384; at 832×480, it requires 18.1 GB memory and and a runtime of 40s. These results indicate that, although LayerT2V has a longer wall-clock runtime, its throughput is comparable after normalizing by total generated outputs.
>
> | Method | Resolution | Output | Peak Memory | Runtime (s) | Effective Output FPS |
> |---|---:|---|---:|---:|---:|
> | LayerT2V | 672×384 | 4 aligned layers, **161 total frames** | 24.4 GB | 157 | **1.03** |
> | Wan 2.1 1.3B | 832×480 | 1 video, **41 total frames** | 18.1 GB | 40 | **1.03** |

---

> > ### Author Rebuttal · Reviewer_pNzn · 2026-04-02
> >
> > The rebuttal satisfies my suggestions/concerns, so I keep my score.

---

> > > ### Author Response · Authors · 2026-04-04
> > >
> > > Thank you again for your helpful feedback and careful review. We sincerely appreciate your time, consideration, and valuable insights, and we are glad that our clarifications have addressed your concerns. We would be grateful if you might kindly consider the possibility of increasing the rating.
> > >
> > > If there are any remaining issues that we have not yet fully addressed, we would appreciate the opportunity to clarify them further.

---

### Decision · Program_Chairs · 2026-04-30

**Decision:**

Accept (regular)

**Comment:**

The final ratings are mixed (1 Weak Reject, 2 Weak Accept, 1 Accept), but overall feedback is positive. Before the rebuttal, the main concern was limited novelty, with some reviewers seeing the method as incremental. Other concerns included weak analysis of scalability (e.g., long videos and multiple subjects), limited ablations, unclear dataset quality control, and insufficient evaluation of editing and failure cases. The rebuttal addressed most of these issues by clarifying temporal consistency, computational cost, dataset validation, and the roles of different modules. During the discussion, reviewer pNzn agreed that it is reasonable to accept the paper, leading to a consensus to acceptance.